# Support Recovery for Orthogonal Matching Pursuit: Upper and Lower bounds

**Raghav Somani**[*]
Microsoft Research, India
t-rasom@microsoft.com

**Chirag Gupta**[* †]
Machine Learning Department,
Carnegie Mellon University
chiragg@andrew.cmu.edu

**Prateek Jain**
Microsoft Research, India
prajain@microsoft.com

**Praneeth Netrapalli**
Microsoft Research, India
praneeth@microsoft.com

## Abstract

We study the problem of sparse regression where the goal is to learn a sparse vector that best optimizes a given objective function. Under the assumption that the objective function satisfies restricted strong convexity (RSC), we analyze orthogonal matching pursuit (OMP), a greedy algorithm that is used heavily in applications, and obtain a support recovery result as well as a tight generalization error bound for the OMP estimator. Further, we show a lower bound for OMP, demonstrating that both our results on support recovery and generalization error are tight up to logarithmic factors. To the best of our knowledge, these are the first such tight upper and lower bounds for *any* sparse regression algorithm under the RSC assumption.

## 1 Introduction

The goal in sparse regression is to find the optimal sparse vector that minimizes a given objective function. Sparse regression is an important problem in statistical machine learning since sparse models lead to better generalization guarantees when the feature dimension is high or data is less, eg, high-dimensional statistics [19], bioinformatics [18], etc. Sparse models also have a smaller memory footprint and are thus useful for resource-constrained machine learning [9]. For simplicity of exposition, we focus on the problem of sparse *linear* regression (SLR), which is a representative problem in this domain. Results for this problem typically extend easily to the general case. Given $\mathbf{A} \in \mathbb{R}^{n \times d}$ and $\mathbf{y}$, the goal of SLR is to recover a sparse vector $\bar{\mathbf{x}}$ that minimizes $\|\mathbf{A}\mathbf{x} - \mathbf{y}\|_2^2$.

The unconditional version of sparse regression can be shown to be NP-hard via a reduction to 3-set cover [14]. However, the problem has been studied heavily under a variety of assumptions such as incoherence [7], null-space property [8], restricted isometry property (RIP) or restricted strong convexity (RSC) [4, 15]. RSC, in particular, is one of the weakest and most popular assumptions for sparse regression problems and has been studied in the context of various algorithms [27, 11, 1, 13]. In this paper, we study the SLR problem under RSC condition.

Typically SLR is studied with one of two goals: a) support recovery, i.e., recovering support (or features) of $\bar{\mathbf{x}}$ and b) bounding generalization error $\left(\|\mathbf{A}(\mathbf{x} - \bar{\mathbf{x}})\|_2^2/n\right)$ which bounds excess error on unseen test points if each row of $\mathbf{A}$ is sampled from a fixed distribution. In general, support recovery

---

[*]Equal contribution
[†]Work done in part while Chirag Gupta was a Research Fellow at Microsoft Research, India

is a more fundamental and challenging problem as a strong support recovery result usually tends to provide strong generalization error bound.

Existing sparse regression algorithms can be broadly categorized into three categories: a) $\ell_1$ minimization or LASSO based algorithms [6, 5, 1], b) non-convex penalty based methods [2, 11, 13], c) greedy methods [22, 17, 27, 12]. In this work, we focus on OMP which is a greedy method that incrementally adds elements to support based on the amount of reduction in training error. Owing to its simplicity, flexibility, and strong practical performance, OMP is one of the most celebrated and practically used algorithms for sparse regression.

OMP has been shown to provide support recovery in noiseless settings, i.e., when $\mathbf{y} = \mathbf{A}\bar{\mathbf{x}}$, under various conditions like incoherence [8], null-space property, RIP/RSC [28] etc. In the noisy setting, while the generalization error of OMP has been studied [28] under RSC, these bounds do not match known lower bounds [29] in terms of the restricted strong convexity constant. In fact, the tightest known generalization error upper bound for polynomial time algorithms is a factor of restricted strong convexity constant worse than the known lower bound [29, 28, 11, 30]. Furthermore, strong support recovery results under RSC are also known only for a non-convex SCAD/MCP penalty based method [13]. For greedy methods, there have been several recent works [20, 21, 25] that consider the problem of support recovery. However, none of these works give strong results for this problem.

In this work, we significantly improve upon these support recovery results for OMP. We show that if the smallest element of $\bar{\mathbf{x}}$ is larger than an appropriate noise level, then OMP recovers the *full* support of $\bar{\mathbf{x}}$ (see Theorem 3.1). As noted in remarks 3 and 4 below the theorem, our result has a better dependence on the restricted condition number than the ones in [20, 21, 25]. The proof of Theorem 3.1 exploits the fact that if a certain element of $\bar{\mathbf{x}}$ is not included in the current support set, then a single step of OMP should lead to a large *additive* decrease in the error. In addition, we present a generalization error analysis for OMP.

Finally, we provide matching lower bounds for our support recovery and generalization error results. To this end, we construct a design matrix that ensures that OMP picks incorrect indices until a large number of elements are added to the support set (see Theorems 4.2, 4.3). As the support set size has to increase arbitrarily for recovery, this also implies poor generalization error (see Theorem 4.3).

We note that our lower bound results are unconditional and are directly applicable to OMP. In contrast, existing lower bounds such as [29] obtain a lower bound for generalization error of *any* polynomial time algorithm assuming $\mathbf{NP} \not\subset \mathbf{P/poly}$. Moreover, these lower bound results are restricted to algorithms which recover *exactly* $s^*$-sparse vectors, where $s^* = |\text{supp}(\bar{\mathbf{x}})|$ and hence do not apply to OMP if it adds more than $s^*$ elements to the support set, which is the more meaningful scenario to consider. Moreover, if each element of $\bar{\mathbf{x}}$ is *large*, then the claim of [29] is almost vacuous as one can recover the support exactly which is the main problem in SLR. In that case, while the generalization error lower bound of [29] holds, it does not preclude the OMP algorithm from recovering the correct support (see Section 4).

**Notation:** Matrices are typically written in bold capital letters (such as $\mathbf{A}$ and $\mathbf{\Sigma}$), vectors are typically written in bold small case letters (such as $\mathbf{x}$ and $\boldsymbol{\eta}$) and universal constants independent of problem parameters are written as $C_1$, $C_2$, etc. For a matrix $\mathbf{A}$, $\mathbf{A}_i$ represents its $i^{th}$ column and $\mathbf{A_S}$ represent the sub-matrix of $\mathbf{A}$ with columns in the index set $\mathbf{S}$. $\rho_s^+(\mathbf{A}^T\mathbf{A})$, $\rho_s^-(\mathbf{A}^T\mathbf{A})$ are restricted smoothness and restricted strong convexity constants of the matrix $\mathbf{A}$ (defined below). $\widetilde{\kappa}_s(\mathbf{A}^T\mathbf{A}) := \rho_1^+(\mathbf{A}^T\mathbf{A})/\rho_s^-(\mathbf{A}^T\mathbf{A})$ for all $s > 0$. $\rho_s^+$, $\rho_s^-$ and $\widetilde{\kappa}_s$ when used without parameter, represent $\rho_s^+(\mathbf{A}^T\mathbf{A})$, $\rho_s^-(\mathbf{A}^T\mathbf{A})$ and $\widetilde{\kappa}_s(\mathbf{A}^T\mathbf{A})$ respectively. The non-zero element of $\bar{\mathbf{x}}$ with the least absolute value is denoted as $\bar{x}_{\min}$.

## 2  Preliminaries and Setting

In this section, we will present some preliminaries and the problem setting considered in this paper. Broadly, we are interested in sparse estimation problems where we are given a function $Q(\cdot)$ and we wish to solve $\min_{\mathbf{x}:\|\mathbf{x}\|_0 \leq s^*} Q(\mathbf{x})$. This problem is in general NP-hard even when $Q(\cdot)$ is a quadratic function. So, we consider this problem under *restricted strong convexity (RSC)* and *restricted smoothness (RSS)* assumptions. While part of our results apply to this general setting, for simplicity of presentation, we focus on the case where $Q(\cdot)$ is a quadratic. More concretely, in the sparse linear regression problem where we are given a measurement matrix $\mathbf{A} \in \mathbb{R}^{n \times d}$ and response $\mathbf{y} \in \mathbb{R}^n$

---
**Algorithm 1** Orthogonal Matching Pursuit (OMP)
---
1: **procedure** OMP($s$)
2:      $\mathbf{S}_0 = \phi$, $\mathbf{x}_0 = \mathbf{0}$, $\mathbf{r}_0 = \mathbf{y}$
3:      **for** $k = 1, 2, \ldots, s$ **do**
4:          $j \leftarrow \underset{i \notin \mathbf{S}_{k-1}}{\arg\max} |\mathbf{A}_i^T \mathbf{r}_{k-1}|$
5:          $\mathbf{S}_k \leftarrow \mathbf{S}_{k-1} \cup \{j\}$
6:          $\mathbf{x}_k \leftarrow \underset{\mathrm{supp}(\mathbf{x}) \subseteq \mathbf{S}_k}{\arg\min} \|\mathbf{A}\mathbf{x} - \mathbf{y}\|_2^2$
7:          $\mathbf{r}_k \leftarrow \mathbf{y} - \mathbf{A}\mathbf{x}_k$
8:      **end for**
9:      **return** $\mathbf{x}_s$
10: **end procedure**
---

and we wish to solve $\min_{\|\mathbf{x}\|_0 \leq s^*} \|\mathbf{A}\mathbf{x} - \mathbf{y}\|_2^2$. We assume that the measurement matrix $\mathbf{A}$ satisfies restricted strong convexity and restricted smoothness properties [4]:

**Definition 2.1** (Restricted strong convexity (RSC)). $\mathbf{A}$ *is said to be restricted strongly convex at level $s$ with parameter $\rho_s^-$ if for every $\mathbf{x}$ and $\mathbf{z}$ such that $\|\mathbf{x} - \mathbf{z}\|_0 \leq s$, we have*

$$\|\mathbf{A}\mathbf{x} - \mathbf{A}\mathbf{z}\|_2^2 \geq \rho_s^- \|\mathbf{x} - \mathbf{z}\|_2^2.$$

**Definition 2.2** (Restricted smoothness (RSS)). $\mathbf{A}$ *is said to be restricted smooth at level $s$ with parameter $\rho_s^+$ if for every $\mathbf{x}$ and $\mathbf{z}$ such that $\|\mathbf{z} - \mathbf{x}\|_0 \leq s$, we have*

$$\|\mathbf{A}\mathbf{x} - \mathbf{A}\mathbf{z}\|_2^2 \leq \rho_s^+ \|\mathbf{x} - \mathbf{z}\|_2^2.$$

The above definitions capture the standard strong convexity and smoothness properties but only in sparse directions. Similarly, we can define a notion of *restricted condition number*.

**Definition 2.3** (Restricted condition number). *The restricted condition number at level $s$ of a matrix $\mathbf{A}$ is defined as*

$$\widetilde{\kappa}_s(\mathbf{A}^T\mathbf{A}) = \frac{\rho_1^+}{\rho_s^-}. \tag{2.1}$$

Throughout this paper, we assume that $\mathbf{A}$ satisfies the above properties and denote the corresponding parameters as $\rho_s^-$, $\rho_s^+$, and $\widetilde{\kappa}_s$ respectively. For our lower bound matrices in Section 4 we show that these properties are satisfied.

**Definition 2.4** ($\ell_\infty - norm$). *We define the $\ell_\infty - norm$ of a matrix $\mathbf{A}$:*

$$\|\mathbf{A}\|_\infty := \max_{\|\mathbf{x}\|_\infty = 1} \|\mathbf{A}\mathbf{x}\|_\infty \tag{2.2}$$

We work under the generative model where $\bar{\mathbf{x}}$ is an $s^*$-sparse vector supported on $\mathbf{S}^*$, that generates the data. More concretely, we assume that the measurements $\mathbf{y}$ are generated as noisy linear measurements of $\bar{\mathbf{x}}$:

$$\mathbf{y} = \mathbf{A}\bar{\mathbf{x}} + \boldsymbol{\eta}, \tag{2.3}$$

where each element of $\boldsymbol{\eta}$ is a mean zero sub-Gaussian random variable with parameter $\sigma$. This means that for some constant $C$, we have,

$$P\{|\eta_i| > t\} \leq C \exp\left(-t^2/2\sigma^2\right).$$

The non-zero element of $\bar{\mathbf{x}}$ with the least absolute value is denoted as $\bar{x}_{\min}$.

In this problem setting, there are two critical questions:

1. **Support recovery**: The goal here is to recover the support of $\bar{\mathbf{x}}$ after observing $\mathbf{y}$ and $\mathbf{A}$. This question can also be posed as estimating $\bar{\mathbf{x}}$ in the $\ell_\infty$ norm i.e., find $\hat{\mathbf{x}}$ such that $\|\hat{\mathbf{x}} - \bar{\mathbf{x}}\|_\infty$ is small.

2. **Generalization error**: Here, the goal is to compute an $\hat{\mathbf{x}}$ such that $\|\mathbf{A}(\hat{\mathbf{x}} - \bar{\mathbf{x}})\|_2$ is small. This quantity is essentially the generalization error when the learned $\hat{\mathbf{x}}$ is used to make prediction over test data generated from same distribution as training data $\mathbf{A}$ and $\mathbf{y}$.

Table 1: Comparison between our results and several prior results on support recovery for Sparse Linear Regression. HTP refers to Hard Thresholding Pursuit, PHT refers to Partial Hard Thresholding, and IHT referes to Iterative Hard Thresholding. These are all thresholding based greedy algorithms. Apart from $\widetilde{\kappa}(\cdot)$, we also use $\kappa_s(\cdot) = \rho_s^+(\cdot)/\rho_s^-(\cdot)$. All values are correct upto constants; we have skipped order notation in the interest of succinctness. Support expansion refers to the value of $s$ in the paper. The $|\bar{x}_{\min}|$ column refers to the condition for support recovery guarantee. All support recovery happens with some probability $\delta$, and we incur polynomial factors of $\log(d/\delta)$ in the $|\bar{x}_{\min}|$ condition. We skip these in the interest of succinctness.

| **Related Work** | **Support expansion** $(s)$ | $|\bar{x}_{\min}|$ **lower bound** |
|---|---|---|
| Yuan et al. [25] [HTP] | $\kappa_{2s}^2 s^*$ | $\dfrac{\sigma\sqrt{s}}{\sqrt{\rho_{2s}^-}}$ |
| Shen et al. [20] [HTP] | $\kappa_{2s}^2 s^*$ | $\dfrac{\sigma\sqrt{\kappa_{2s}}\sqrt{\rho_1^+ s}}{\rho_{s+s^*}^-}$ |
| Shen et al. [21] [PHT$(r)$] | $s^* + \kappa_{2s}^2 \min\{s^*, r\}$ | $\dfrac{\sigma\sqrt{\kappa_{2s}}\sqrt{\rho_1^+ s}}{\rho_{2s}^-}$ |
| Jain et al. [11] [IHT] | $\kappa_{2s+s^*}^2 s^*$ | $-$ |
| Zhang [28] [OMP] | $\widetilde{\kappa}_{s+s^*} s^* \log \kappa_{s+s^*}$ | $-$ |
| Theorem 3.1 [OMP] | $\widetilde{\kappa}_{s+s^*} s^* \log \kappa_{s+s^*}$ | $\gamma \cdot \dfrac{\sigma\sqrt{\rho_1^+}}{\rho_{s+s^*}^-}$ |

We note that in both the above problems we are allowed to output $\hat{x}$ that may have $s \geq s^*$ elements in the support. This is a standard and crucial relaxation needed to provide strong guarantees under weak assumptions for SLR. This work considers orthogonal matching pursuit (OMP) [16, 23] for solving both of the above problems. OMP is one of the most popular methods for sparse optimization and it is essentially a greedy method that incrementally estimates the support of $\bar{x}$ by adding one element at a time. See Algorithm 1 for a pseudo-code of OMP for SLR.

In Section 3 we show our upper bounds for the performance of OMP with respect to both the problems above, under the RSS/RSC conditions. In Section 4, we provide a matching lower bound (upto logarithmic factors) which shows that there exist certain sparse linear regression problems on which OMP cannot perform significantly better than the error bounds given by our analysis. In Section 5 we show some simple simulations to ground our results.

## 3 Upper bounds for OMP

We first present our key contribution which is a support recovery bound for OMP under RSC/RSS.

**Theorem 3.1** (Support Recovery for OMP). *Let* $\mathbf{A} \in \mathbb{R}^{n \times d}$ *and* $\bar{\mathbf{x}} \in \mathbb{R}^d$ *be a* $s^*$-*sparse vector. Let* $\mathbf{y} = \mathbf{A}\bar{\mathbf{x}} + \boldsymbol{\eta}$ *and let* $\hat{\mathbf{x}}_s$ *be the output of OMP after* $s$ *iterations, where*

$$s \geq C_1 \widetilde{\kappa}_{s+s^*} s^* \cdot \log\left(\frac{5\rho_{s+s^*}^+}{\rho_{s+s^*}^-}\right),$$

*and* $\widetilde{\kappa}_{s+s^*}$ *is the restricted condition number (Definition 2.3). Moreover, let* $\left\|\mathbf{A}_{\mathbf{S}^*\backslash\mathbf{S}}^T \mathbf{A}_{\mathbf{S}} (\mathbf{A}_{\mathbf{S}}^T \mathbf{A}_{\mathbf{S}})^{-1}\right\|_\infty \leq \gamma$ *where* $\mathbf{S} = supp(\hat{\mathbf{x}}_s)$. *Then, for every* $\delta \in \left(0, e^{-68}\right)$, *if*

$$|\bar{x}_{\min}| \geq \left(1 + \sqrt{2}\left(1 + \gamma\right)\right) \frac{\sigma}{\rho_{s+s^*}^-} \sqrt{\rho_1^+ \log \frac{d}{\delta}}, \qquad (3.1)$$

*and* $s + s^* \geq \log\left(1/\delta\right)$, *then* $\mathbf{S}^* \subseteq supp(\hat{\mathbf{x}}_s)$ *and* $\|\hat{\mathbf{x}}_s - \bar{\mathbf{x}}\|_\infty \leq \sigma\sqrt{\frac{2}{\rho_s^-} \log\left(s/\delta\right)}$ *with probability at least* $1 - 7\delta$. *Here* $C_1 = 664$ *is a universal constant.*

**Remark 1**: $\rho_{s+s^*}^-$ is the RSC constant of the $\|\mathbf{A}\mathbf{x} - \mathbf{y}\|_2^2$ objective. Hence $\rho_{s+s^*}^-$ is $n$ times the restricted strong convexity of the normalized objective $\frac{1}{n}\|\mathbf{A}\mathbf{x} - \mathbf{y}\|_2^2$ whose scale is independent of $n$. Similarly, $\sqrt{\rho_1^+}$ hides a $\sqrt{n}$. Thus $|\bar{x}_{\min}|$ essentially scales as $1/\sqrt{n}$.

**Remark 2**: The $\gamma$ parameter in the above theorem is somewhat similar to the standard incoherence parameter [24], although the incoherence parameter can be significantly larger than $\gamma$. Further, existing results for OMP [26] require the incoherence parameter to be strictly less than 1 while our analysis holds for arbitrary values of $\gamma$. Thus, our results apply to more general design matrices $\mathbf{A}$.
**Remark 3**: Our assumption on $|\bar{x}_{\min}|$ is better at least by a factor of $\sqrt{\tilde{\kappa}}$ than corresponding assumptions made in recent work that analyzes OMP for support recovery [20, 21, 25] (see Table 1).
**Remark 4**: To the best of our knowledge, [13] is the only known support recovery result for LASSO under RSC, that provides strong guarantees as our result above. However, the non-convex penalty based algorithm of [13] might produce iterates which are dense, so intermediate steps can be more expensive than sparsity preserving OMP. Furthermore, while qualitatively, our bound is similar to the bound of [13], their proof requires $n \geq \|\bar{\mathbf{x}}\|_1^2 \log d$ which, naïvely, for many problems with imbalanced non-zero elements of $\bar{\mathbf{x}}$ can be as large as $(s^*)^2$.

**Proof Sketch of Theorem 3.1 (see Appendix B.2 for details):** Theorem 3.2 (stated below) guarantees that OMP has a very small objective value after a certain number of support expansion steps. This guarantees small generalization error (Theorem 3.3), but not support recovery. To guarantee support recovery, our proof critically exploits a novel observation (Lemma B.4 in Appendix B.2) that if at any iteration of OMP, full support recovery has not happened, then OMP decreases function value by a fixed, but small, additive constant. Theorem 3.2 allows us to say that even this small constant decrement cannot happen for too long since the objective value is already small. Overall, this means that support recovery must happen soon after we have small objective value.

Let $s$ be the iteration index that is sufficient to satisfy the conditions for Theorem 3.2. From Theorem 3.2 we have with probability at least $1 - 2\delta$,

$$\|\mathbf{A}\mathbf{x}_s - \mathbf{y}\|_2^2 \leq \|\mathbf{A}\bar{\mathbf{x}} - \mathbf{y}\|_2^2 + 40\frac{\sigma^2 s \rho_1^+ \log(d/\delta)}{\rho_{s+s^*}^+}. \tag{3.2}$$

$$\leq \|\boldsymbol{\eta}\|_2^2 + 40\sigma^2 s \log(d/\delta)$$

Suppose any one of the support index has not been recovered (that is, $|\mathbf{S}^* \backslash \mathbf{S}| > 0$) then if $j \in (\mathbf{S}^* \backslash \mathbf{S})^c$ is selected by OMP in its $(s+1)^{th}$ iteration, we have by step 4 of Algorithm 1,

$$\left\| \mathbf{A}_{\mathbf{S}^* \backslash \mathbf{S}}^T \mathbf{r}_s \right\|_\infty \leq |\mathbf{A}_j^T \mathbf{r}_s|. \tag{3.3}$$

In Lemma B.4, we lower bound the LHS of (3.3) as follows:

$$\left\| \mathbf{A}_{\mathbf{S}^* \backslash \mathbf{S}}^T \mathbf{r}_s \right\|_\infty \geq \rho_{s+s^*}^- |\bar{x}_{\min}| - \sqrt{2}(1+\gamma)\sigma\sqrt{\rho_1^+ \log(d/\delta)}, \tag{3.4}$$

with probability at least $1 - 2\delta$. Since $|\bar{x}_{\min}| \geq \left(1 + \sqrt{2}(1+\gamma)\right)\frac{\sigma}{\rho_{s+s^*}^-}\sqrt{\rho_1^+ \log(d/\delta)}$, combining (3.3) with (3.4) gives,

$$\sigma^2 \log\frac{d}{\delta} \leq \frac{1}{\rho_1^+}\left(\mathbf{A}_j^T \mathbf{r}_s\right)^2. \tag{3.5}$$

This gives us an additive decrease in the function value:

$$\|\mathbf{A}\mathbf{x}_{s+1} - \mathbf{y}\|_2^2 \leq \min_{x_j}\|\mathbf{A}_j x_j - \mathbf{r}_s\|_2^2$$

$$= \|\mathbf{A}\mathbf{x}_s - \mathbf{y}\|_2^2 - \frac{1}{\rho_1^+}\left(\mathbf{A}_j^T \mathbf{r}_s\right)^2 \leq \|\mathbf{A}\mathbf{x}_s - \mathbf{y}\|_2^2 - \sigma^2 \log(d/\delta) \tag{3.6}$$

Suppose that for another $l$ iterations, the full support is not recovered. Then,

$$\|\mathbf{A}\mathbf{x}_{s+l} - \mathbf{y}\|_2^2 \leq \|\mathbf{A}\mathbf{x}_s - \mathbf{y}\|_2^2 - \sigma^2 l \log(d/\delta). \tag{3.7}$$

Further it can be shown that the function value at iteration $s + l$ cannot be too small,

$$\|\mathbf{A}\mathbf{x}_{s+l} - \mathbf{y}\|_2^2 \geq \|\boldsymbol{\eta}\|_2^2 - \sigma^2(s + l + s^*) - 4\sigma^2(s + l + s^*)\sqrt{\log(d/\delta)}, \tag{3.8}$$

with probability at least $1 - \delta$. Therefore combining (3.8) and (3.2) and plugging them in (3.7), we finally get,

$$l \leq 80s + s + s^* = \mathcal{O}(s). \tag{3.9}$$

Therefore with good probability, OMP recovers the full support in $\mathcal{O}(s)$ iterations. See Appendix B.2 for details. $\qquad\square$

We now bound the training error for OMP after running a certain number of iterations (which are fewer than the number of iterations required for support recovery as shown in Theorem 3.1). The proof of this theorem follows via a modification of the proof of Lemma A.5 in [28]. See Appendix B.1 for the proof.

**Theorem 3.2** (Training Error for OMP). *Consider the setting of Theorem 3.1. Also, let*

$$s \geq 8\widetilde{\kappa}_{s+s^*}s^* \cdot \log\left(\frac{5\rho^+_{s+s^*}}{\rho^-_{s+s^*}}\right).$$

*Then with probability $1-2\delta$, the output $\hat{\mathbf{x}}_s$ of OMP after $s$ steps satisfies:*

$$\frac{1}{n}\left\|\mathbf{A}\hat{\mathbf{x}}_s - \mathbf{y}\right\|_2^2 \leq \frac{1}{n}\left\|\mathbf{A}\bar{\mathbf{x}} - \mathbf{y}\right\|_2^2 + 40\frac{\sigma^2 s \log(d/\delta)}{\rho^+_{s+s^*}} \cdot \frac{\rho^+_1}{n}. \tag{3.10}$$

Given good objective value decrease, we can show a tight generalization error on the output of OMP. While in general support recovery is the main goal of a sparse regression algorithm, in several problem scenarios one might not care about support recovery and focus only on the accuracy of the learned predictor. See Appendix B.3 for the proof.

**Theorem 3.3** (Generalization Error for OMP). *Consider the setting of Theorem 3.1. Let $\hat{\mathbf{x}}_s$ be the output of OMP after $s$ iterations. For any constant $C_1 \geq 8$, there exists a constant $C_2(\leq 9C_1)$ such that if $s$ satisfies,*

$$C_1\widetilde{\kappa}_{s+s^*}s^* \cdot \log\left(\frac{5\rho^+_{s+s^*}}{\rho^-_{s+s^*}}\right) \geq s \geq 8\widetilde{\kappa}_{s+s^*}s^* \cdot \log\left(\frac{5\rho^+_{s+s^*}}{\rho^-_{s+s^*}}\right),$$

*then with probability at least $1 - 4\delta$,*

$$\frac{1}{n}\left\|\mathbf{A}(\hat{\mathbf{x}}_s^{OMP} - \bar{\mathbf{x}})\right\|_2^2 \leq C_2 \frac{\sigma^2\widetilde{\kappa}_{s+s^*}s^*}{n} \cdot \log\left(\frac{5\rho^+_{s+s^*}}{\rho^-_{s+s^*}}\right) \cdot \log\frac{d}{\delta}. \tag{3.11}$$

## 3.1 Gaussian ensemble

Finally, we instantiate the above theorems for a Gaussian ensemble, i.e., when $\mathbf{A}$ is sampled from a Gaussian distribution $\mathcal{N}(\mathbf{0}, \boldsymbol{\Sigma})$. We denote the maximum and the minimum singular values of $\boldsymbol{\Sigma}$ as $\sigma_{\max}$ and $\sigma_{\min}$ and the condition number of $\boldsymbol{\Sigma}$ as $\kappa(\boldsymbol{\Sigma})$. To the best of our knowledge, the following is the best known generalization error guarantee in this setting in terms of the dependence on $\kappa(\boldsymbol{\Sigma})$.

**Corollary 3.3.1** (Gaussian ensemble: generalization error). *Let the rows of the matrix $\mathbf{A} \in \mathbb{R}^{n \times d}$ be sampled from $\mathcal{N}(\mathbf{0}, \boldsymbol{\Sigma})$ where $\Sigma_{ii} \leq 1 \ \forall \ i \in [d]$ and $\bar{\mathbf{x}}$ be a $s^*$-sparse vector. Let $\hat{\mathbf{x}}_s$ be the output of OMP after $s$ iterations and $\mathbf{S} = supp(\hat{\mathbf{x}}_s)$ be the support recovered, where,*

$$s = C_2\kappa(\boldsymbol{\Sigma}) \cdot \log\left(45\kappa(\boldsymbol{\Sigma})\right)s^*, n > 4C_1\frac{s\log d}{\sigma_{\min}(\boldsymbol{\Sigma})}, \text{ and } s + s^* \geq \log\frac{1}{\delta},$$

*for any $\delta > 0$. Then with probability at least $1 - 4\delta - e^{-C_0 n}$, the following holds:*

$$\frac{1}{n}\left\|\mathbf{A}(\hat{\mathbf{x}}_s^{OMP} - \bar{\mathbf{x}})\right\|_2^2 \leq C_3 \frac{\sigma^2\kappa(\boldsymbol{\Sigma})s^*}{n} \cdot \log\left(45\kappa(\boldsymbol{\Sigma})\right) \cdot \log\frac{d}{\delta}$$

*Here $C_0, C_1, C_3$ and $C_4$ are universal constants independent of any problem parameters.*

Note the linear dependence of generalization error on $\kappa(\boldsymbol{\Sigma})$. This matches the lower bound of [29], although technically the bound does not apply to OMP as $s > s^*$. The proof follows directly from Theorem 3.3 along with standard concentration results. See Appendix B.3 for details.

We now present support recovery result for Gaussian ensembles. For simplicity, we consider the case when $\mathbf{A}$ is sampled from $\mathcal{N}(\mathbf{0}, \mathbf{I})$. This can also be extended to $\mathcal{N}(\mathbf{0}, \boldsymbol{\Sigma})$ but involves cumbersome linear algebraic computations, which we avoid for simplicity.

**Corollary 3.3.2** (Gaussian ensemble: support recovery)**.** *Let the rows of the matrix* $\mathbf{A} \in \mathbb{R}^{n \times d}$ *be sampled from* $\mathcal{N}(\mathbf{0}, \mathbf{I}_{d \times d})$ *and* $\bar{\mathbf{x}}$ *be a* $s^*$*-sparse vector. Suppose further that* $|\bar{x}_{\min}| \geq 23\sigma\sqrt{\frac{\log(d/\delta)}{n}}$. *Let* $\hat{\mathbf{x}}_s$ *be the output of OMP after* $s$ *iterations and* $\mathbf{S} = supp(\hat{\mathbf{x}}_s)$ *be the support recovered, where,*

$$s \geq C_1 s^*, n > C_2(s^*)^2 \log\frac{d}{\delta}, \text{ and } s + s^* \geq \log\frac{1}{\delta},$$

*for any* $\delta > 0$. *Then* $\mathbf{S}^* \subseteq supp(\hat{\mathbf{x}}_s)$ *and* $\|\hat{\mathbf{x}}_s - \bar{\mathbf{x}}\|_{\infty} \leq 2\sigma\sqrt{\frac{2\log(s/\delta)}{n}}$ *with probability at least* $1 - e^{-C_0 n} - 9\delta$. *Here* $C_0, C_1$ *and* $C_2$ *are universal constants independent of any problem parameter.*

This matches the bounds of [13] up to constants. The proof directly follows from Theorem 3.1 along with standard Gaussian concentration results. See Appendix B.3 for details.

## 4 Lower bounds for OMP

In this section, we provide lower bounds on the performance of OMP, both in terms of support recovery and generalization error. These bounds show that:

- The imperative quantities we make assumptions on in the upper bound section, viz: $\widetilde{\kappa}_{s+s^*}$ and $\gamma$ are relevant and meaningful.
- Given bounds on these quantities, our results are tight, up to logarithmic factors.

To provide these lower bounds, we construct matrices $\mathbf{M}^{(\epsilon)}$ that are parametrized by $\epsilon$. We fix $\bar{\mathbf{x}}$ to be an $s^*$-sparse vector such that:

$$\begin{cases} \bar{x}_i = \sqrt{1/s^*} & \text{if } 1 \leq i \leq s^*, \\ \bar{x}_i = 0 & \text{if } s^* < i. \end{cases} \tag{4.1}$$

Thus, $\mathbf{S}^* := supp(\bar{\mathbf{x}}) = \{1, 2, \ldots, s^*\}$. All our lower bound theorems use this fixed vector which is independent of the noise level $\sigma$. Our results are thus stronger than a typical minimax rate in which $\bar{\mathbf{x}}$ can be *scaled* based on $\sigma$. For instance, the lower bounds of [29], [30] use such a strategy. Also, the support is distributed evenly across the $\bar{x}_i$'s (4.1). Thus, we show that even large elements are not recovered.

We now define $\mathbf{M}^{(\epsilon)} \in \mathbb{R}^{n \times d}$ for a given $\epsilon \in [0, 1]$, any $s^* \leq d \leq n$ in the following manner: $\mathbf{M}^{(\epsilon)}_{1:s^*}$ are random orthogonal vectors such that $\left\|\mathbf{M}^{(\epsilon)}_i\right\|^2_2 = n$, $\forall i \in [s^*]$. For $i \in [d] \setminus [s^*]$, each column vector is defined as,

$$\mathbf{M}^{(\epsilon)}_i = \sqrt{\frac{1-\epsilon}{s^*}} \sum_{j=1}^{s^*} \mathbf{M}^{(\epsilon)}_j + \sqrt{\epsilon}\, \mathbf{g}_i, \tag{4.2}$$

where $\mathbf{g}_i$ is such that $\|\mathbf{g}_i\|^2_2 = n$, $\mathbf{g}_i^T \mathbf{M}^{(\epsilon)}_{1:s^*} = \mathbf{0}$ and $\mathbf{g}_i^T \mathbf{g}_j = 0$ for all $i \neq j$.

The intuition behind this construction is that OMP would prefer the *average* direction $\mathbf{M}^{(\epsilon)}_{\mathbf{S}^*}\bar{\mathbf{x}}$ over any of the correct directions $\mathbf{M}^{(\epsilon)}_i$, where $i \in \mathbf{S}^*$. Thus, we add a scaled version of $\mathbf{M}^{(\epsilon)}_{\mathbf{S}^*}\bar{\mathbf{x}}$ to each of the other orthogonal vectors of the matrix.

The parameter $\epsilon$ is set carefully to ensure that the condition number of the matrix does not increase too much, so that $\mathbf{M}^{(\epsilon)}$ satisfies the constraints of Theorem 3.3 and Theorem 3.1 (upto constants). This is captured in the next lemma:

**Lemma 4.1.** *The matrix* $\mathbf{M}^{(\epsilon)}$ *satisfies*

- $\widetilde{\kappa}_s\left(\mathbf{M}^{(\epsilon)}\right) \leq 4(1 + 2(1 - \epsilon)s) = \mathcal{O}(s)$

- $\left\|\mathbf{M}^{(\epsilon)T}_{\mathbf{S}^* \setminus \mathbf{S}}\mathbf{M}^{(\epsilon)}_{\mathbf{S}}\left(\mathbf{M}^{(\epsilon)T}_{\mathbf{S}}\mathbf{M}^{(\epsilon)}_{\mathbf{S}}\right)^{-1}\right\|_{\infty} \leq \frac{1}{\sqrt{s^*(1-\epsilon)}}$ *for* $\mathbf{S} \cap \mathbf{S}^* = \phi$.

We now use the above construction to show that in the noiseless case, i.e., when $\mathbf{y} = \mathbf{M}^{(\epsilon)}\bar{\mathbf{x}}$, OMP fails to recover any of the support elements in $\mathbf{S}^*$ for some $\epsilon$. Similarly, we show that in the noisy case, support recovery fails and hence the generalization error of OMP is also large and matches the upper bound provided in Theorem 3.3. Proofs for this section can be found in Appendix C.

### 4.1 Noiseless case

For the deterministic noiseless case, i.e., $\sigma = 0$, we consider the matrix $\mathbf{M}^{(\epsilon)}$ for $\epsilon = (1 - 3/2s^*)$ and show that OMP requires to add *all* the elements in support to recover the correct support.

**Theorem 4.2.** *For every value of $d, n$ and $s^*$ where $s^* \leq d \leq n$, there exists a design matrix $\mathbf{A} \in \mathbb{R}^{n \times d}$ and a $s^*$-sparse vector $\bar{\mathbf{x}}$ (defined in (4.2), (4.1)) such that the following holds true for OMP when applied to the sparse linear regression problem with $\mathbf{y} = \mathbf{A}\bar{\mathbf{x}}$ and when OMP is executed for $s \leq d - s^*$ iterations:*

- $\widetilde{\kappa}_s(\mathbf{A}) \leq 16(s/s^*)$ *and* $\gamma \leq \sqrt{2/3}$.

- *The support set $\mathbf{S}$ recovered by OMP after $s$ iterations is disjoint from $\mathbf{S}^*$, i.e., $\mathbf{S}^* \cap \mathbf{S} = \phi$.*

Our support recovery result in Theorem 3.1 requires $s \geq C\widetilde{\kappa}_{s+s^*} s^*$ and one natural question is whether running OMP for $s$ iterations is necessary for recovering the actual support. This theorem guarantees that it is indeed the case, i.e., if design matrix $\mathbf{A}$ is ill-conditioned then OMP has to work with support sets of size $s \geq \widetilde{\kappa}_{s+s^*} s^*$. This in turn implies that the number of rows in $\mathbf{A}$ (i.e., sample complexity) should also scale with $\widetilde{\kappa}_{s+s^*}$.

Note that the lower bound results of [29], [30] do not provide any insights for how the sample complexity of an algorithm should scale with $\kappa_{s+s^*}$ for support recovery. In fact for this problem their results are vacuous if $|\bar{x}_{\min}|$ is reasonably large. For instance, with the $\bar{\mathbf{x}}$ defined in (4.1) and the design matrix proposed by [29], OMP can recover the true support of $\bar{\mathbf{x}}$ *exactly* after just $\mathcal{O}(s^*)$ iterations with $n = s^* \log d$ samples. Thus, a large condition number of $\mathbf{A}$ in their construction does not imply difficulty in recovery for OMP.

### 4.2 Noisy case

For the noisy case, i.e., $\sigma \neq 0$, we can study both support recovery as well as generalization error behavior with respect to the restricted condition number $\widetilde{\kappa}_{s+s^*}$. For this section, we consider the matrix $\mathbf{M}^{(\epsilon)}$ for $\epsilon = (1 - 1/4s^*)$. That is, we show that with high probability, OMP starts recovering the correct support only after $d^{1-\alpha}$ iterations for some constant $\alpha > 0$. This further implies that the generalization error cannot be better than the lower bound on generalization error we showed in Theorem 3.3 (upto constants).

**Theorem 4.3.** *For every value of $d$ and $s^*$, and any constants $\alpha \in (0,1)$, $\delta \in (0,1)$, such that $8 \leq s^* \leq s \leq d^{1-\alpha}$ and $d \geq \max\left\{32 \log(1/\delta), 4^{1/\alpha}\right\}$, there exists a sparse linear regression problem with $\mathbf{y} = \mathbf{A}\bar{\mathbf{x}} + \boldsymbol{\eta}$, $\boldsymbol{\eta} \sim \mathcal{N}(\mathbf{0}, \sigma^2 \mathbf{I}_{n \times n})$, with design matrix $\mathbf{A}$, and a $s^*$-sparse vector $\bar{\mathbf{x}}$ defined in (4.2),(4.1) such that the following holds:*

- $\widetilde{\kappa}_s(\mathbf{A}) \leq 36(s/s^*)$ *for all $s$ and* $\gamma \leq 1/2$,

- *With probability at least $1 - \delta$, the output $\hat{\mathbf{x}}_s$ of OMP after $s$ steps satisfies:*

$$\frac{1}{n}\|\mathbf{A}\hat{\mathbf{x}}_s - \mathbf{A}\bar{\mathbf{x}}\|_2^2 \geq \frac{\sigma^2 \widetilde{\kappa}_{s+s^*} s^*}{18n} \cdot \log \frac{d}{\delta},$$

- *Support set $\mathbf{S}$ recovered by OMP after $s$ iterations is disjoint from $\mathbf{S}^*$.*

Note that the dependence of the generalization error bound on $\widetilde{\kappa}_{s+s^*}$ matches our generalization error bound in Theorem 3.3. Interestingly, for our construction, noise ends up helping recovery because while Theorem 4.2 ensures that the recovery of true support elements does not occur till the very last step, noise can only help in recovering one of the true elements. However, the probability of picking up the correct element by chance is tiny as we restrict $s \leq d^{1-\alpha}$. We in fact believe that the result holds generally for any $s$ and $d$. However, proving it turns out to be quite intricate since it requires finer results about the the behavior of the order statistics of independent Gaussian variables.

## 5 Simulations

In this section, we present simulations that verify our results. In particular, we generate a matrix $\mathbf{M}^{(\epsilon)} \in \mathbb{R}^{1000 \times 100}$, and a fixed $s^* = 10$-sparse vector $\bar{\mathbf{x}}$ by using the construction given in (4.2) and

(4.1) where $\epsilon \in (0,1)$. We then generate $\mathbf{y} = \mathbf{M}^{(\epsilon)}\bar{\mathbf{x}} + \sigma\mathcal{N}(\mathbf{0}, \mathbf{I}_{n \times n})$ and apply OMP for recovering the support of $\bar{\mathbf{x}}$. $\hat{s}(\epsilon)$ denotes the index or support set size that is needed by OMP for fully recovering the support of $\bar{\mathbf{x}}$.

Note that we can also compute the actual value of $\widetilde{\kappa}$ for $\mathbf{M}^{(\epsilon)}$; in general the restricted condition number of $\mathbf{M}^{(\epsilon)}$ increases with decreasing $\epsilon$, thus increasing the difficulty of the support recovery problem.

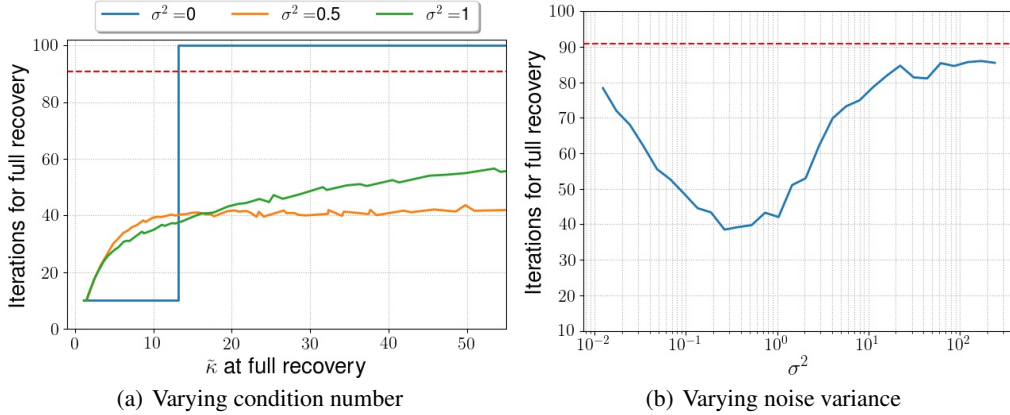

(a) Varying condition number　　　　　(b) Varying noise variance

Figure 1: Number of iterations required for recovering the full support of $\bar{\mathbf{x}}$ with respect to the restricted condition number ($\widetilde{\kappa}_{s+s^*}$) of the design matrix and the sub-Gaussian parameter of the noise term ($\sigma^2$).

Figure 1(a) plots $\hat{s}(\epsilon)$ (i.e. support size required for full recovery) against restricted condition number $\widetilde{\kappa}(\mathbf{M}^{(\epsilon)})$ of $\mathbf{M}^{(\epsilon)}$ generated by varying $\epsilon \in (0,1)$. Theorem 4.2 claims that for $\sigma = 0$, full recovery requires $\widetilde{\kappa}_s$ to be smaller than $\mathcal{O}(d/s^*)$, which is observed in Figure 1(a). For larger variance $\sigma^2$, full recovery requires larger number of iterations for smaller $\widetilde{\kappa}$.

As mentioned in the remark below Theorem 4.3, adding noise can only help in case of large $\widetilde{\kappa}$ as our construction precludes full recovery unless $s = d$. We observe this behavior in both Figure 1(a) and 1(b), where slightly larger value of $\sigma$ ends up helping support recovery, but for larger values of noise variance, OMP's performance is similar to an algorithm that simply selects each feature uniformly at random.

## 6　Conclusion

In this paper, we analyze OMP for the sparse regression problem under RSC/RSS assumptions. We obtain support recovery and generalization guarantees for OMP under this setting. We also provide lower bounds for OMP showing that our results are tight up to logarithmic factors. We note that our results significantly improve upon a long list of existing results for greedy methods and match the best known results for sparse regression that use nonconvex penalty based methods. In contrast to nonconvex penalty methods however, OMP guarantees the sparsity of intermediate iterates and hence can be much more efficient. We also verify our results with synthetic experiments.

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
