[Supplementary Material]

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

# A  General properties of Gaussians

**Lemma A.1.** *If $\mathbf{U} \in \mathbb{R}^{n \times k}$ has unit norm orthogonal columns where $k \geq \log \frac{1}{\delta}$, and $\boldsymbol{\eta} \sim \mathcal{N}(\mathbf{0}, \sigma^2 \mathbf{I}_{n \times n})$ then*

$$\left\| \mathbf{U}^T \boldsymbol{\eta} \right\|_2^2 \leq \sigma^2 k + 4\sigma^2 \sqrt{k \log \frac{1}{\delta}} \tag{A.1}$$

*holds with probability at least $1 - \delta$*

*Proof.* Using an exponential probability tail inequality of positive semidefinite quadratic forms in a sub-gaussian random vector [10],

$$P \left\{ \frac{1}{\sigma^2} \left\| \mathbf{U}^T \boldsymbol{\eta} \right\|_2^2 > Tr(\mathbf{U}\mathbf{U}^T) + 2\sqrt{Tr(\mathbf{U}\mathbf{U}^T\mathbf{U}\mathbf{U}^T)t} + 2 \left\| \mathbf{U} \right\|_2 t \right\} \leq e^{-t}$$

$$\text{or, } P \left\{ \frac{1}{\sigma^2} \left\| \mathbf{U}^T \boldsymbol{\eta} \right\|_2^2 > k + 2\sqrt{kt} + 2t \right\} \leq e^{-t}$$

Setting $t = \log \frac{1}{\delta}$ and assuming $k \geq \log \frac{1}{\delta}$, we get the required result. $\qquad\square$

# B  Proofs of results in Section 3

---
**Algorithm 2** Orthogonal Matching Pursuit (OMP) for General $Q(\cdot)$
---
1: **procedure** GENERAL OMP($s$)
2: $\quad$ $\mathbf{S}_0 = \phi, \mathbf{x}_0 = \mathbf{0}$
3: $\quad$ **for** $k = 1, 2, \ldots, s$ **do**
4: $\quad\quad$ $j \leftarrow \underset{i \notin \mathbf{S}_{k-1}}{\arg\max} |\nabla Q(\mathbf{x}_{k-1})|$
5: $\quad\quad$ $\mathbf{S}_k \leftarrow \mathbf{S}_{k-1} \cup \{j\}$
6: $\quad\quad$ $\mathbf{x}_k \leftarrow \underset{\text{supp}(\mathbf{x}) \subseteq \mathbf{S}_k}{\arg\min} \; Q(\mathbf{x})$
7: $\quad$ **end for**
8: $\quad$ **return** $\mathbf{x}_s$
9: **end procedure**

---

## B.1  Proof of Theorem 3.2

We wish to prove Theorem 3.2 in this section. We in fact prove a more general version of the theorem that holds for any function $Q(\cdot)$. Algorithm 2 is the OMP algorithm generalized to any $Q(\cdot)$. To show guarantees in this setting, we assume that $Q(\cdot)$ satisfies Restricted Smoothness (RSS) and Restricted Strong Convexity (RSC) properties, given by respective constants $\rho_s^+$ and $\rho_s^-$ parametrized by the sparsity level $s$. We need that for all $\mathbf{x}, \mathbf{y}$ such that $\|\mathbf{y} - \mathbf{x}\|_0 \leq s$,

$$\rho_s^- \|\mathbf{y} - \mathbf{x}\|_2^2 \leq Q(\mathbf{y}) - Q(\mathbf{x}) - \langle \nabla Q(\mathbf{x}), \mathbf{y} - \mathbf{x} \rangle \leq \rho_s^+ \|\mathbf{y} - \mathbf{x}\|_2^2 . \tag{B.1}$$

Note that for the sparse linear regression problem, setting the objective function to $Q(\mathbf{x}) := \|\mathbf{A}\mathbf{x} - \mathbf{y}\|_2^2$ converts Algorithm 2 to Algorithm 1. Further, this $Q(\cdot)$ indeed satisfies the RSS and RSC constants of the matrix $\mathbf{A}$ as defined in definitions 2.2 and 2.1.

In order to present this result, we need the following notation from [28]. For any sparsity level $s$ and any vector $\mathbf{x}$, define

$$\epsilon_s(\mathbf{x}) = \sup_{\|\mathbf{u}\|_0 = s, \|\mathbf{u}\|_2 = 1} |\nabla Q(\mathbf{x})^T \mathbf{u}|, \tag{B.2}$$

$\epsilon_s(\mathbf{x})$ is essentially the $\ell_2$ norm of $\nabla Q(\mathbf{x})$ restricted to its largest $s$ coordinates (in absolute value). We are specifically interested in $\epsilon_s(\bar{\mathbf{x}})$ where $\bar{\mathbf{x}}$ is an $s^*$-sparse vector that we are trying to estimate. In particular for the linear regression case, $\bar{\mathbf{x}}$ is the generative parameter ($\mathbf{y} = \mathbf{A}\bar{\mathbf{x}} + \boldsymbol{\eta}$). As we shall see in B.3, $\epsilon_s(\bar{\mathbf{x}})$ determines the sub-optimality of the OMP estimator on the sparse linear regression problem. Thus, we first write a lemma quantifying how large this quantity can be.

**Lemma B.1.** *For the sparse linear regression problem with $Q(\mathbf{x}) = \|\mathbf{A}\mathbf{x} - \mathbf{y}\|_2^2$, suppose that $\max_i \|\mathbf{A}_i\|_2^2 \leq n$. Then with probability at least $1 - \delta$,*

$$\epsilon_s^2(\bar{\mathbf{x}}) \leq 8\sigma^2 \rho_1^+ s \log \frac{2d}{\delta}. \tag{B.3}$$

*Proof.* For the sparse linear regression problem,

$$
\begin{aligned}
\epsilon_s(\bar{\mathbf{x}}) &= \sup_{\|\mathbf{u}\|_0 = s, \|\mathbf{u}\|_2 = 1} \left| \nabla Q(\bar{\mathbf{x}})^T \mathbf{u} \right| \\
&= \sup_{\|\mathbf{u}\|_0 = s, \|\mathbf{u}\|_2 = 1} \left| 2\mathbf{u}^T \mathbf{A}^T \boldsymbol{\eta} \right| \\
&\leq \sup_{\|\mathbf{u}\|_0 = s, \|\mathbf{u}\|_2 = 1} \left( 2 \|\mathbf{u}\|_2 \left\| (\mathbf{A}^T \boldsymbol{\eta})_{\mathrm{supp}(\mathbf{u})} \right\|_2 \right) \\
&= 2 \sup_{|\mathbf{S}| = s} \left\| \mathbf{A}_{\mathbf{S}}^T \boldsymbol{\eta} \right\|_2 \\
&\leq 2\sqrt{s} \left\| \mathbf{A}^T \boldsymbol{\eta} \right\|_\infty \\
&\overset{\xi_1}{\leq} 2\sqrt{s}\sigma \sqrt{2\rho_1^+ \log \frac{2d}{\delta}} \\
&= 2\sqrt{2}\sigma \sqrt{\rho_1^+ s \log \frac{2d}{\delta}} \\
\implies \epsilon_s^2(\bar{\mathbf{x}}) &\leq 8\sigma^2 \rho_1^+ s \log \frac{2d}{\delta}
\end{aligned}
$$

Here $\xi_1$ holds with probability $(1 - \delta)$. □

We now write the key technical lemma, which is a strengthened version of Lemma A.5 from [28]. We show that if OMP is run to a slightly larger (only in constants) expansion set then we get much better convergence to $Q(\bar{\mathbf{x}})$. This improved convergence is key to achieving a better generalization error (Theorem 3.3). For the statement and proof of the following lemma, we use the notation of [28]. Additionally, we define $s^* = |\bar{F}|$.

**Lemma B.2** (Modified version of lemma A.5 in [28]). *Suppose OMP is run till $s \geq k + s^*$ steps, where*

$$k \geq 4|\bar{F} \setminus F^{(0)}| \frac{\rho_1^+}{\rho_s^-} \log \left( 20 \left( \frac{\rho_s^+}{\rho_s^-} \right)^2 \right), \text{ then}$$

$$Q(\mathbf{x}_k) \leq Q(\bar{\mathbf{x}}) + 2.5\epsilon_s(\bar{\mathbf{x}})^2 / \rho_s^+.$$

*Proof.* We strengthen the proof of Lemma A.5 of [28] appropriately. First, create the split on $L$ differently, using the following $\mu$:

$$\mu = 10 \left( \frac{\rho_s^+}{\rho_s^-} \right)^2. \tag{B.4}$$

This value of $\mu$ will be plugged in later.

Next, we use lemma A.1 from [28] at level $s$ instead of level $m$ as has been done in the original proof. This leads to a modified value of $q_l$:

$$\min_{\mathbf{x} \in \bar{F}_l} Q(\mathbf{x}) \leq Q(\bar{\mathbf{x}}) + q_l, \text{ where}$$

$$q_l = 1.5\rho_s^+ \sum_{i=2^l}^m \bar{x}_i^2 + 0.5 \frac{\epsilon_s(\bar{\mathbf{x}})^2}{\rho_s^+}.$$

Thus, equation (12) in [28] becomes

$$Q(\mathbf{x}_k) - Q(\bar{\mathbf{x}}) \leq 3\mu^{-1}\rho_s^+ \sum_{i=2^l}^m \bar{x}_i^2 + \frac{0.5}{\rho_s^+}(1 + \mu^{-1})\epsilon_s(\bar{\mathbf{x}})^2. \tag{B.5}$$

We have a different splitting condition compared to [28]. Now, either

$$2\mu^{-1}\rho_s^+ \sum_{i=2^l}^{m} \bar{x}_i^2 \leq \frac{1}{\rho_s^+}(1 + \mu^{-1})\epsilon_s(\bar{\mathbf{x}})^2$$

in which case the theorem follows by simply replacing the first term in the RHS of equation B.5 and using $(1 + \mu^{-1}) \leq 2$. We consider the other case:

$$2\mu^{-1}\rho_s^+ \sum_{i=2^l}^{m} \bar{x}_i^2 > \frac{1}{\rho_s^+}(1 + \mu^{-1})\epsilon_s(\bar{\mathbf{x}})^2.$$

In this case, we use the same approach as in [28], but modify from the fourth inequality onwards (see below),

$$\rho_s^- \|\mathbf{x}_k - \bar{\mathbf{x}}\|_2^2 \leq 2(Q(\mathbf{x}_k) - Q(\bar{\mathbf{x}})) + \frac{\epsilon_s(\bar{\mathbf{x}})^2}{\rho_s^-}$$

$$\leq 6\mu^{-1}\rho_s^+ \sum_{i=2^l}^{m} \bar{x}_i^2 + (2 + \mu^{-1})\frac{\epsilon_s(\bar{\mathbf{x}})^2}{\rho_s^-}$$

$$\leq 6\mu^{-1}\rho_s^+ \sum_{i=2^l}^{m} \bar{x}_i^2 + 2(1 + \mu^{-1})\frac{\epsilon_s(\bar{\mathbf{x}})^2}{\rho_s^-}$$

$$\leq 6\mu^{-1}\rho_s^+ \sum_{i=2^l}^{m} \bar{x}_i^2 + \left(\rho_s^+ \sum_{i=2^l}^{m} \bar{x}_i^2\right)\frac{4\rho_s^+\mu^{-1}}{\rho_s^-}$$

$$= \left(\frac{4\rho_s^+\mu^{-1}}{\rho_s^-} + 6\mu^{-1}\right)\rho_s^+ \sum_{i=2^l}^{m} \bar{x}_i^2$$

$$\leq \left(\frac{10\rho_s^+\mu^{-1}}{\rho_s^-}\right)\rho_s^+ \sum_{i=2^l}^{m} \bar{x}_i^2$$

$$\overset{\xi_1}{=} \rho_s^- \sum_{i=2^l}^{m} \bar{x}_i^2.$$

In $\xi_1$ we plug in the value of $\mu$ from B.4. The rest of the proof follows in the same manner as the original proof. □

**Theorem B.3.** *If $Q(\cdot)$ satisfies RSC and RSS and if*

$$s \geq 8\widetilde{\kappa}_{s+s^*}s^* \log 5\kappa_{s+s^*}, \tag{B.6}$$

*then the output $\hat{\mathbf{x}}_s$ of OMP after $s$ steps satisfies*

$$Q(\hat{\mathbf{x}}_s) \leq Q(\bar{\mathbf{x}}) + 2.5\frac{\epsilon_{s+s^*}(\bar{\mathbf{x}})^2}{\rho_{s+s^*}^+}. \tag{B.7}$$

*Proof.* Writing Lemma B.2 other words, if OMP is run till $\bar{s} \geq k + s^*$ steps, where

$$k \geq 8\widetilde{\kappa}_{\bar{s}}s^* \log 5\kappa_{\bar{s}}$$

then,

$$Q(\mathbf{x}_k) \leq Q(\bar{\mathbf{x}}) + 2.5\frac{\epsilon_{\bar{s}}(\bar{\mathbf{x}})^2}{\rho_{\bar{s}}^+}.$$

Setting $k = s$ and $\bar{s} = s + s^*$, we have that if OMP is run till $s + s^*$ steps, where

$$s \geq 8s^*\widetilde{\kappa}_{s+s^*} \log 5\kappa_{s+s^*}$$

then,

$$Q(\mathbf{x}_s) \leq Q(\bar{\mathbf{x}}) + 2.5\frac{\epsilon_{s+s^*}(\bar{\mathbf{x}})^2}{\rho_{s+s^*}^+}$$

□

*Proof of Theorem 3.2.* We set $Q(\mathbf{x}) = \|\mathbf{A}\mathbf{x} - \mathbf{y}\|_2^2$ in Theorem B.3. Now, plugging in the value of $\epsilon_s(\bar{\mathbf{x}})^2$ from Lemma B.1, we get that with probability at least $1 - 2\delta$,

$$\|\mathbf{A}\hat{\mathbf{x}}_s - \mathbf{y}\|_2^2 \leq \|\mathbf{A}\bar{\mathbf{x}} - \mathbf{y}\|_2^2 + 2.5\frac{8\sigma^2\rho_1^+(s + s^*)}{\rho_{s+s^*}^+}\log\frac{d}{\delta}$$

$$\implies \frac{1}{n}\|\mathbf{A}\hat{\mathbf{x}}_s - \mathbf{y}\|_2^2 \leq \frac{1}{n}\|\mathbf{A}\bar{\mathbf{x}} - \mathbf{y}\|_2^2 + 40\frac{\sigma^2\rho_1^+ s\log(d/\delta)}{n\rho_{s+s^*}^+}.$$

$\square$

## B.2 Proof of Theorem 3.1

We first prove the following crucial lemma required for Theorem 3.1, as also discussed in the main paper.

**Lemma B.4.** *Let $\mathbf{A} \in \mathbb{R}^{n \times d}$ and $\bar{\mathbf{x}} \in \mathbb{R}^d$ be a $s^*$-sparse vector. Let $\mathbf{y} = \mathbf{A}\bar{\mathbf{x}} + \boldsymbol{\eta}$ and let $\hat{\mathbf{x}}_s$ be the output of OMP after $s$ iterations, where*

$$s \geq 8\widetilde{\kappa}_{s+s^*}s^* \cdot \log\left(\frac{5\rho_{s+s^*}^+}{\rho_{s+s^*}^-}\right),$$

*and $\widetilde{\kappa}_{s+s^*}$ is the restricted condition number (Definition 2.3). Moreover, let $\left\|\mathbf{A}_{\mathbf{S}^*\backslash\mathbf{S}}^T\mathbf{A}_\mathbf{S}(\mathbf{A}_\mathbf{S}^T\mathbf{A}_\mathbf{S})^{-1}\right\|_\infty \leq \gamma$ where $\mathbf{S} = supp(\hat{\mathbf{x}}_s)$. Then, if*

$$|\bar{x}_{\min}| \geq (1 + \sqrt{2}(1 + \gamma))\frac{\sigma}{\rho_{s+s^*}^-}\sqrt{\rho_1^+\log d/\delta}, \tag{B.8}$$

*then for all $l \geq 0$, if complete support recovery hasn't happened till $s + l$ iterations of OMP, then with high probability, the function value after $s + l$ iterations satisfies,*

$$\|\mathbf{A}\mathbf{x}_{s+l} - \mathbf{y}\|_2^2 \leq \|\mathbf{A}\mathbf{x}_s - \mathbf{y}\|_2^2 - \sigma^2 l\log\frac{d}{\delta} \tag{B.9}$$

*Proof.* If OMP at $(s+1)^{th}$ picks up an index $j$ in $(\mathbf{S}^*)^c$, then

$$\left\|\mathbf{A}_{\mathbf{S}^*\backslash\mathbf{S}}^T\mathbf{A}_\mathbf{S}^\perp(\mathbf{A}\bar{\mathbf{x}} + \boldsymbol{\eta})\right\|_\infty \leq |\mathbf{A}_j^T\mathbf{A}_\mathbf{S}^\perp(\mathbf{A}\bar{\mathbf{x}} + \boldsymbol{\eta})| \tag{B.10}$$

We now compute a lower bound for the L.H.S. in (B.10):

$$\left\|\mathbf{A}_{\mathbf{S}^*\backslash\mathbf{S}}^T\mathbf{A}_\mathbf{S}^\perp(\mathbf{A}\bar{\mathbf{x}} + \boldsymbol{\eta})\right\|_\infty \geq \left\|\mathbf{A}_{\mathbf{S}^*\backslash\mathbf{S}}^T\mathbf{A}_\mathbf{S}^\perp\mathbf{A}_{\mathbf{S}^*}\bar{\mathbf{x}}_{\mathbf{S}^*}\right\|_\infty - \left\|\mathbf{A}_{\mathbf{S}^*\backslash\mathbf{S}}^T\mathbf{A}_\mathbf{S}^\perp\boldsymbol{\eta}\right\|_\infty$$

$$= \left\|\mathbf{A}_{\mathbf{S}^*\backslash\mathbf{S}}^T\mathbf{A}_\mathbf{S}^\perp\mathbf{A}_{\mathbf{S}^*\backslash\mathbf{S}}\bar{\mathbf{x}}_{\mathbf{S}^*\backslash\mathbf{S}}\right\|_\infty - \left\|\mathbf{A}_{\mathbf{S}^*\backslash\mathbf{S}}^T\mathbf{A}_\mathbf{S}^\perp\boldsymbol{\eta}\right\|_\infty$$

$$\geq \frac{\left\|\mathbf{A}_{\mathbf{S}^*\backslash\mathbf{S}}^T\mathbf{A}_\mathbf{S}^\perp\mathbf{A}_{\mathbf{S}^*\backslash\mathbf{S}}\bar{\mathbf{x}}_{\mathbf{S}^*\backslash\mathbf{S}}\right\|_2}{\sqrt{|\mathbf{S}^*\backslash\mathbf{S}|}} - \left\|\mathbf{A}_{\mathbf{S}^*\backslash\mathbf{S}}^T\mathbf{A}_\mathbf{S}^\perp\boldsymbol{\eta}\right\|_\infty$$

$$\geq \frac{\rho^-(\mathbf{A}_{\mathbf{S}^*\backslash\mathbf{S}}^T\mathbf{A}_\mathbf{S}^\perp\mathbf{A}_{\mathbf{S}^*\backslash\mathbf{S}})\left\|\bar{\mathbf{x}}_{\mathbf{S}^*\backslash\mathbf{S}}\right\|_2}{\sqrt{|\mathbf{S}^*\backslash\mathbf{S}|}} - \left\|\mathbf{A}_{\mathbf{S}^*\backslash\mathbf{S}}^T\mathbf{A}_\mathbf{S}^\perp\boldsymbol{\eta}\right\|_\infty$$

$$\geq \rho^-(\mathbf{A}_{\mathbf{S}^*\backslash\mathbf{S}}^T\mathbf{A}_\mathbf{S}^\perp\mathbf{A}_{\mathbf{S}^*\backslash\mathbf{S}})|\bar{x}_{\min}| - \left\|\mathbf{A}_{\mathbf{S}^*\backslash\mathbf{S}}^T\mathbf{A}_\mathbf{S}^\perp\boldsymbol{\eta}\right\|_\infty \tag{B.11}$$

We can analyze the first term in (B.11) as follows. Consider the matrix,

$$\mathbf{A}_{\mathbf{S}\cup\mathbf{S}^*}^T\mathbf{A}_{\mathbf{S}\cup\mathbf{S}^*} = \begin{bmatrix}\mathbf{A}_{\mathbf{S}^*\backslash\mathbf{S}} & \mathbf{A}_\mathbf{S}\end{bmatrix}^T\begin{bmatrix}\mathbf{A}_{\mathbf{S}^*\backslash\mathbf{S}} & \mathbf{A}_\mathbf{S}\end{bmatrix} = \begin{bmatrix}\mathbf{A}_{\mathbf{S}^*\backslash\mathbf{S}}^T\mathbf{A}_{\mathbf{S}^*\backslash\mathbf{S}} & \mathbf{A}_{\mathbf{S}^*\backslash\mathbf{S}}^T\mathbf{A}_\mathbf{S} \\ \mathbf{A}_\mathbf{S}^T\mathbf{A}_{\mathbf{S}^*\backslash\mathbf{S}} & \mathbf{A}_\mathbf{S}^T\mathbf{A}_\mathbf{S},\end{bmatrix}$$

and its Schur complement with respect to its block $\mathbf{A}_\mathbf{S}^T\mathbf{A}_\mathbf{S}$:

$$\begin{bmatrix}\mathbf{A}_{\mathbf{S}\cup\mathbf{S}^*}^T\mathbf{A}_{\mathbf{S}\cup\mathbf{S}^*}/\mathbf{A}_\mathbf{S}^T\mathbf{A}_\mathbf{S}\end{bmatrix} = \mathbf{A}_{\mathbf{S}^*\backslash\mathbf{S}}^T\mathbf{A}_{\mathbf{S}^*\backslash\mathbf{S}} - \mathbf{A}_{\mathbf{S}^*\backslash\mathbf{S}}^T\mathbf{A}_\mathbf{S}(\mathbf{A}_\mathbf{S}^T\mathbf{A}_\mathbf{S})^{-1}\mathbf{A}_\mathbf{S}^T\mathbf{A}_{\mathbf{S}^*\backslash\mathbf{S}}$$

$$= \mathbf{A}_{\mathbf{S}^*\backslash\mathbf{S}}^T(\mathbf{I} - \mathbf{A}_\mathbf{S}(\mathbf{A}_\mathbf{S}^T\mathbf{A}_\mathbf{S})^{-1}\mathbf{A}_\mathbf{S}^T)\mathbf{A}_{\mathbf{S}^*\backslash\mathbf{S}}$$

$$= \mathbf{A}_{\mathbf{S}^*\backslash\mathbf{S}}^T\mathbf{A}_\mathbf{S}^\perp\mathbf{A}_{\mathbf{S}^*\backslash\mathbf{S}} \tag{B.12}$$

Therefore from the Schur eigenvalue inequality, we have

$$\rho^- \left( \left[ \mathbf{A}_{\mathbf{S} \cup \mathbf{S}^*}^T \mathbf{A}_{\mathbf{S} \cup \mathbf{S}^*} / \mathbf{A}_{\mathbf{S}}^T \mathbf{A}_{\mathbf{S}} \right] \right) \geq \rho^- (\mathbf{A}_{\mathbf{S} \cup \mathbf{S}^*}^T \mathbf{A}_{\mathbf{S} \cup \mathbf{S}^*}) \geq \rho_{s+s^*}^- \tag{B.13}$$

We can upper bound the second term in (B.11) by using Cauchy-Schwarz and directly assuming an infinity norm upper bound as

$$
\begin{aligned}
\left\| \mathbf{A}_{\mathbf{S}^* \setminus \mathbf{S}}^T \mathbf{A}_{\mathbf{S}}^{\perp} \boldsymbol{\eta} \right\|_{\infty} &= \left\| \mathbf{A}_{\mathbf{S}^* \setminus \mathbf{S}}^T (\mathbf{I} - \mathbf{A}_{\mathbf{S}} (\mathbf{A}_{\mathbf{S}}^T \mathbf{A}_{\mathbf{S}})^{-1} \mathbf{A}_{\mathbf{S}}^T) \boldsymbol{\eta} \right\|_{\infty} \\
&\leq \left\| \mathbf{A}_{\mathbf{S}^* \setminus \mathbf{S}}^T \boldsymbol{\eta} \right\|_{\infty} + \left\| \mathbf{A}_{\mathbf{S}^* \setminus \mathbf{S}}^T \mathbf{A}_{\mathbf{S}} (\mathbf{A}_{\mathbf{S}}^T \mathbf{A}_{\mathbf{S}})^{-1} \mathbf{A}_{\mathbf{S}}^T \boldsymbol{\eta} \right\|_{\infty} \\
&\leq \left\| \mathbf{A}_{\mathbf{S}^* \setminus \mathbf{S}}^T \boldsymbol{\eta} \right\|_{\infty} + \left\| \mathbf{A}_{\mathbf{S}^* \setminus \mathbf{S}}^T \mathbf{A}_{\mathbf{S}} (\mathbf{A}_{\mathbf{S}}^T \mathbf{A}_{\mathbf{S}})^{-1} \right\|_{\infty} \left\| \mathbf{A}_{\mathbf{S}}^T \boldsymbol{\eta} \right\|_{\infty} \\
&\leq (1 + \gamma) \left\| \mathbf{A}_{\mathbf{S} \cup \mathbf{S}^*}^T \boldsymbol{\eta} \right\|_{\infty} \qquad \left( \text{Assuming } \left\| \mathbf{A}_{\mathbf{S}^* \setminus \mathbf{S}}^T \mathbf{A}_{\mathbf{S}} (\mathbf{A}_{\mathbf{S}}^T \mathbf{A}_{\mathbf{S}})^{-1} \right\|_{\infty} \leq \gamma \right) \\
&\leq \sqrt{2} (1 + \gamma) \sigma \sqrt{\rho_1^+ \log \frac{2(s + s^*)}{2\delta}} \qquad \text{(with probability at least } 1 - 2\delta) \\
&\leq \sqrt{2} (1 + \gamma) \sigma \sqrt{\rho_1^+ \log \frac{d}{\delta}} \qquad \text{(using } d \geq s + s^*)
\end{aligned}
\tag{B.14}
$$

Using (B.13) and (B.14) in (B.11), we have,

$$\left\| \mathbf{A}_{\mathbf{S}^* \setminus \mathbf{S}}^T \mathbf{A}_{\mathbf{S}}^{\perp} (\mathbf{A}\bar{\mathbf{x}} + \boldsymbol{\eta}) \right\|_{\infty} \geq \rho_{s+s^*}^- |\bar{x}_{\min}| - \sqrt{2}(1 + \gamma) \sigma \sqrt{\rho_1^+ \log \frac{d}{\delta}} \tag{B.15}$$

Since $|\bar{x}_{\min}| \geq (1 + \sqrt{2}(1 + \gamma)) \frac{\sigma}{\rho_{s+s^*}^-} \sqrt{\rho_1^+ \log d/\delta}$, from Equation (B.10) and (B.15) we get

$$
\begin{aligned}
|\mathbf{A}_j^T \mathbf{r}_s| &= |\mathbf{A}_j^T \mathbf{A}_{\mathbf{S}}^{\perp} (\mathbf{A}\bar{\mathbf{x}} + \boldsymbol{\eta})| \\
&\geq \rho_{s+s^*}^- |\bar{x}_{\min}| - \sqrt{2}(1 + \gamma) \sigma \sqrt{\rho_1^+ \log \frac{d}{\delta}} \\
&\geq \sigma \sqrt{\rho_1^+ \log \frac{d}{\delta}} \\
\implies \sigma^2 \log \frac{d}{\delta} &\leq \frac{1}{\rho_1^+} \left( \mathbf{A}_j^T \mathbf{r}_s \right)^2
\end{aligned}
\tag{B.16}
$$

Now, we upper bound the function value at iteration $s + 1$:

$$
\begin{aligned}
\|\mathbf{A}\mathbf{x}_{s+1} - \mathbf{y}\|_2^2 &\leq \min_{x_j} \|\mathbf{A}_j x_j - \mathbf{r}_s\|_2^2 \\
&= \left\| \mathbf{A}_j (\mathbf{A}_j^T \mathbf{A}_j)^{-1} \mathbf{A}_j^T \mathbf{r}_s - \mathbf{r}_s \right\|_2^2 \\
&= \left\| \frac{1}{\|\mathbf{A}_j\|_2^2} \mathbf{A}_j \mathbf{A}_j^T \mathbf{r}_s - \mathbf{r}_s \right\|_2^2 \\
&= \left\| \frac{1}{\|\mathbf{A}_j\|_2^2} \mathbf{A}_j \mathbf{A}_j^T \mathbf{r}_s \right\|_2^2 + \|\mathbf{r}_s\|_2^2 - 2 \left\langle \mathbf{r}_s, \frac{1}{\|\mathbf{A}_j\|_2^2} \mathbf{A}_j \mathbf{A}_j^T \mathbf{r}_s \right\rangle \\
&= \|\mathbf{A}\mathbf{x}_s - \mathbf{y}\|_2^2 - \frac{1}{\|\mathbf{A}_j\|_2^2} \left( \mathbf{A}_j^T \mathbf{r}_s \right)^2 \\
&\leq \|\mathbf{A}\mathbf{x}_s - \mathbf{y}\|_2^2 - \frac{1}{\rho_1^+} \left( \mathbf{A}_j^T \mathbf{r}_s \right)^2 \\
&\leq \|\mathbf{A}\mathbf{x}_s - \mathbf{y}\|_2^2 - \sigma^2 \log \frac{d}{\delta} \qquad \text{(Using (B.16))}
\end{aligned}
\tag{B.17}
$$

The bound (B.17) and the argument holds for all $t \geq s$ as long as there is at least one more element that OMP has not yet recovered. We wish to show an upper bound for the total number of steps that

OMP can run without recovering all the elements in the support. Suppose OMP runs for another $l$ iterations without recovering all elements in the support. Then, telescoping (B.17), we get

$$\|\mathbf{A}\mathbf{x}_t - \mathbf{y}\|_2^2 - \|\mathbf{A}\mathbf{x}_{t+1} - \mathbf{y}\|_2^2 \geq \sigma^2 \log \frac{d}{\delta} \qquad \forall s \leq t \leq s + l - 1$$

$$\implies \|\mathbf{A}\mathbf{x}_s - \mathbf{y}\|_2^2 - \|\mathbf{A}\mathbf{x}_{s+l} - \mathbf{y}\|_2^2 \geq \sigma^2 l \log \frac{d}{\delta}.$$

$\square$

*Proof of Theorem 3.1.* From Theorem 3.2, for $\bar{s} \geq 8\widetilde{\kappa}_{\bar{s}+s^*} s^* \cdot \log\left(\frac{5\rho_{\bar{s}+s^*}^+}{\rho_{\bar{s}+s^*}^-}\right)$, we have with probability at least $1 - 2\delta$,

$$\|\mathbf{A}\mathbf{x}_{\bar{s}} - \mathbf{y}\|_2^2 \leq \|\boldsymbol{\eta}\|_2^2 + 40\sigma^2 \frac{\bar{s}\rho_1^+}{\rho_{\bar{s}+s^*}^+} \log \frac{d}{\delta} \qquad (\text{B.18})$$

Using Lemma B.4, after $\bar{s} + l$ OMP iterations, if there is always at least one element that is not recovered, the function value will decrease significantly, i.e.,

$$\|\mathbf{A}\mathbf{x}_{\bar{s}} - \mathbf{y}\|_2^2 - \|\mathbf{A}\mathbf{x}_{\bar{s}+l} - \mathbf{y}\|_2^2 \geq \sigma^2 l \log \frac{d}{\delta} \qquad (\text{B.19})$$

We will now lower bound $\|\mathbf{A}\mathbf{x}_{\bar{s}+l} - \mathbf{y}\|_2^2$ showing that $l$ cannot be too large. This is basically asking how best can a sub-Gaussian noise vector be fitted in the $\ell_2$ norm. That is equivalent to the problem

$$\min_{\|\mathbf{x}\|_0 = k} \|\mathbf{A}\mathbf{x} - \boldsymbol{\eta}\|_2^2 = \left\|\mathbf{A}_k^\perp \boldsymbol{\eta}\right\|_2^2$$

$$= \|\boldsymbol{\eta}\|_2^2 - \boldsymbol{\eta}^T \mathbf{A}_k (\mathbf{A}_k^T \mathbf{A}_k)^{-1} \mathbf{A}_k^T \boldsymbol{\eta} \qquad (\text{B.20})$$

$\mathbf{A}_k (\mathbf{A}_k^T \mathbf{A}_k)^{-1} \mathbf{A}_k^T$ is nothing but a projection matrix which can be written as $\mathbf{U}_k \mathbf{U}_k^T$ for some matrix $\mathbf{U}_k$ of size $n \times k$ with unit norm orthogonal columns. Consider any fixed matrix $\mathbf{U}_k'$ of size $n \times k$. Since $k \geq \log \frac{1}{\delta}$ using A.1 we have

$$\boldsymbol{\eta}^T \mathbf{U}_k' \mathbf{U}_k'^T \boldsymbol{\eta} - \sigma^2 k \leq 4\sigma^2 \sqrt{k} \sqrt{\log \frac{1}{\delta}} \qquad (\text{B.21})$$

with probability at least $1 - \delta$. Setting $\delta' = \frac{\delta}{d^k}$,

$$\boldsymbol{\eta}^T \mathbf{U}_k' \mathbf{U}_k'^T \boldsymbol{\eta} - \sigma^2 k \leq 4\sigma^2 \sqrt{k} \sqrt{\log \frac{d^k}{\delta}}$$

with probability at least $1 - \delta/d^k$. Taking a union bound over all possible projection matrices with respect to a $k$ sized subset of the columns of $\mathbf{A}$ (of which there are about $d^k$), we get,

$$\max_{|\mathbf{U}_k'| = k} \left\{ \boldsymbol{\eta}^T \mathbf{U}_k' \mathbf{U}_k'^T \boldsymbol{\eta} \right\} - \sigma^2 k \leq 4\sigma^2 k \sqrt{\log \frac{d}{\delta}} \qquad (\text{with probability at least } 1 - \delta)$$

$$\implies \min_{\|\mathbf{x}\|_0 = k} \|\mathbf{A}\mathbf{x} - \boldsymbol{\eta}\|_2^2 \geq \|\boldsymbol{\eta}\|_2^2 - \sigma^2 k - 4\sigma^2 k \sqrt{\log \frac{d}{\delta}} \qquad (\text{with probability at least } 1 - \delta)$$

$$(\text{B.22})$$

$$\therefore \|\mathbf{A}\mathbf{x}_{\bar{s}+l} - \mathbf{y}\|_2^2 = \|\mathbf{A}\mathbf{x}_{\bar{s}+l} - \mathbf{A}\mathbf{x}^* - \boldsymbol{\eta}\|_2^2$$

$$\geq \min_{\|\mathbf{x}\|_0 = \bar{s}+l+s^*} \|\mathbf{A}\mathbf{x} - \boldsymbol{\eta}\|_2^2$$

$$\geq \|\boldsymbol{\eta}\|_2^2 - \sigma^2(\bar{s} + l + s^*) - 4\sigma^2(\bar{s} + l + s^*) \sqrt{\log \frac{d}{\delta}} \qquad (\text{B.23})$$

Using the upper bound of $\|\mathbf{A}\mathbf{x}_{\bar{s}} - \mathbf{y}\|_2^2$ from (B.18) and lower bound of $\|\mathbf{A}\mathbf{x}_{\bar{s}+l} - \mathbf{y}\|_2^2$ from (B.23) in (B.19) we finally get

$$l\sigma^2 \log \frac{d}{\delta} \leq 40 \frac{\sigma^2 \rho_1^+}{\rho_{\bar{s}+s^*}^+} s \log \frac{d}{\delta} + \sigma^2(\bar{s}+l+s^*) + 4\sigma^2(\bar{s}+l+s^*)\sqrt{\log \frac{d}{\delta}} \quad \text{(w.p. at least } 1-5\delta)$$

$$\implies l \leq 40\bar{s}\frac{\rho_1^+}{\rho_{\bar{s}+s^*}^+} + (\bar{s}+l+s^*)\left[\frac{1}{\log d/\delta} + \frac{4}{\sqrt{\log d/\delta}}\right]$$

$$\implies l \leq \frac{40\bar{s}\frac{\rho_1^+}{\rho_{\bar{s}+s^*}^+}}{1-u} + (\bar{s}+s^*)\frac{u}{1-u} \tag{B.24}$$

where $u := \frac{1}{\log d/\delta} + \frac{4}{\sqrt{\log d/\delta}}$. Since $\log(d/\delta) > 68$, $\frac{1}{1-u} < 2$ and $\frac{u}{1-u} < 1$, and we have

$$l \leq 80\bar{s}\frac{\rho_1^+}{\rho_{\bar{s}+s^*}^+} + \bar{s} + s^* \leq 82\bar{s} \tag{B.25}$$

Thus OMP can run only for another $l$ steps, after which we are ensured recovery. Along with the initial $\bar{s}$ steps, this means that after a total of $s = l + \bar{s} \leq 83\bar{s} = 664\widetilde{\kappa}_{s+s^*}s^* \cdot \log\left(\frac{5\rho_{s+s^*}^+}{\rho_{s+s^*}^-}\right)$ steps, we are ensured complete recovery. This completes the support recovery part of the proof.

We will now show that given full support recovery, i.e., $\mathbf{S}^* \subseteq \mathbf{S}$, $\|\hat{\mathbf{x}}_s - \bar{\mathbf{x}}\|_\infty$ is bounded.

$$\|\hat{\mathbf{x}}_s - \bar{\mathbf{x}}\|_\infty = \left\|\left(\mathbf{A}_{\mathbf{S}}^T \mathbf{A}_{\mathbf{S}}\right)^{-1}\mathbf{A}_{\mathbf{S}}^T(\mathbf{A}_{\mathbf{S}\cup\mathbf{S}^*}\bar{\mathbf{x}}+\boldsymbol{\eta})-\bar{\mathbf{x}}\right\|_\infty$$

$$= \left\|\left(\mathbf{A}_{\mathbf{S}}^T \mathbf{A}_{\mathbf{S}}\right)^{-1}\mathbf{A}_{\mathbf{S}}^T(\mathbf{A}_{\mathbf{S}}\bar{\mathbf{x}}+\boldsymbol{\eta})-\bar{\mathbf{x}}\right\|_\infty$$

$$= \left\|\left(\mathbf{A}_{\mathbf{S}}^T \mathbf{A}_{\mathbf{S}}\right)^{-1}\mathbf{A}_{\mathbf{S}}^T\boldsymbol{\eta}\right\|_\infty \tag{B.26}$$

We will separately analyze the first term in (B.26). Define $\mathbf{X} := \mathbf{A}_{\mathbf{S}}\left(\mathbf{A}_{\mathbf{S}}^T\mathbf{A}_{\mathbf{S}}\right)^{-1}$:

$$\left\|\left(\mathbf{A}_{\mathbf{S}}^T \mathbf{A}_{\mathbf{S}}\right)^{-1}\mathbf{A}_{\mathbf{S}}^T\boldsymbol{\eta}\right\|_\infty = \left\|\mathbf{X}^T\boldsymbol{\eta}\right\|_\infty$$

We will now provide a bound on $\left\|\mathbf{X}^T\boldsymbol{\eta}\right\|_\infty$:

$$P\left\{\left\|\mathbf{X}^T\boldsymbol{\eta}\right\|_\infty > t\right\} \leq \sum_{i=1}^{s} P\left\{\left|\mathbf{X}_i^T\boldsymbol{\eta}\right| > t\right\}$$

$$\leq \sum_{i=1}^{s} 2\exp\left\{-\frac{t^2}{2\sigma^2\|\mathbf{X}_i\|_2^2}\right\}$$

$$\leq 2s\exp\left\{-\frac{t^2}{2\sigma^2 \max_{i\leq s}\|\mathbf{X}_i\|_2^2}\right\} \tag{B.27}$$

Therefore, if $t = \sigma \max_{i\leq s}\|\mathbf{X}_i\|_2 \sqrt{2\log(s/\delta)}$, then

$$P\left\{\left\|\mathbf{X}^T\boldsymbol{\eta}\right\|_\infty \leq \sigma \max_{i\leq s}\|\mathbf{X}_i\|_2 \sqrt{2\log\frac{s}{\delta}}\right\} \geq 1 - 2\delta \tag{B.28}$$

We are now left to bound $\max_{i\leq s}\|\mathbf{X}_i\|_2$ to complete the proof. Note that $\max_{i\leq s}\|\mathbf{X}_i\|_2 \leq \left\|\left(\mathbf{A}_{\mathbf{S}}^T\mathbf{A}_{\mathbf{S}}\right)^{-1}\mathbf{A}_{\mathbf{S}}^T\right\|_2$. Let $\mathbf{w}$ be any non-zero vector in the span of $\mathbf{A}_{\mathbf{S}}$ and define $\mathbf{v} \overset{\text{def}}{=}$

$\left(\mathbf{A_S}^T \mathbf{A_S}\right)^{-1} \mathbf{A_S}^T \mathbf{w}$. Note that

$$\mathbf{A_S v} = \mathbf{A_S} \left(\mathbf{A_S}^T \mathbf{A_S}\right)^{-1} \mathbf{A_S}^T \mathbf{w}$$
$$= \mathbf{w}$$
$$\implies \|\mathbf{w}\|_2 \geq \sqrt{\rho_s^- \left(\mathbf{A_S}^T \mathbf{A_S}\right)} \|\mathbf{v}\|_2$$
$$\text{or, } \frac{\|\mathbf{v}\|_2}{\|\mathbf{w}\|_2} \leq \frac{1}{\sqrt{\rho_s^- \left(\mathbf{A_S}^T \mathbf{A_S}\right)}}$$
$$\implies \max_{i \leq s} \|\mathbf{X}_i\|_2 \leq \frac{1}{\sqrt{\rho_s^- \left(\mathbf{A_S}^T \mathbf{A_S}\right)}} \tag{B.29}$$

Using (B.29) in (B.28) we get

$$P\left\{ \|\mathbf{X}^T \boldsymbol{\eta}\|_\infty \leq \sigma \sqrt{\frac{2}{\rho_s^- \left(\mathbf{A_S}^T \mathbf{A_S}\right)} \log \frac{s}{\delta}} \right\} \geq 1 - 2\delta \tag{B.30}$$

Therefore, we get

$$\|\hat{\mathbf{x}}_s - \bar{\mathbf{x}}\|_\infty \leq \sigma \sqrt{\frac{2}{\rho_s^- \left(\mathbf{A_S}^T \mathbf{A_S}\right)} \log \frac{s}{\delta}} \tag{B.31}$$

$\square$

## B.3    Proof of Theorem 3.3

*Proof of Theorem 3.3.* Let $\mathbf{x} = \hat{\mathbf{x}}_s^{\text{OMP}}$, and $\mathbf{S} = \text{supp}(\mathbf{x})$.

$$\|\mathbf{Ax} - \mathbf{A}\bar{\mathbf{x}}\|_2^2 = \|\mathbf{Ax} - \mathbf{y}\|_2^2 + \|\mathbf{y} - \mathbf{A}\bar{\mathbf{x}}\|_2^2 - 2\langle \mathbf{Ax} - \mathbf{y}, \mathbf{A}\bar{\mathbf{x}} - \mathbf{y}\rangle \tag{B.32}$$
$$= \|\mathbf{Ax} - \mathbf{y}\|_2^2 + \|\boldsymbol{\eta}\|_2^2 + 2\langle \mathbf{Ax} - \mathbf{y}, \boldsymbol{\eta}\rangle$$
$$= \|\mathbf{Ax} - \mathbf{y}\|_2^2 + \|\boldsymbol{\eta}\|_2^2 + 2\langle \mathbf{Ax} - \mathbf{A}\bar{\mathbf{x}} + \mathbf{A}\bar{\mathbf{x}} - \mathbf{y}, \boldsymbol{\eta}\rangle$$
$$= \|\mathbf{Ax} - \mathbf{y}\|_2^2 - \|\boldsymbol{\eta}\|_2^2 + 2\langle \mathbf{Ax} - \mathbf{A}\bar{\mathbf{x}}, \boldsymbol{\eta}\rangle$$
$$\leq \|\mathbf{Ax} - \mathbf{y}\|_2^2 - \|\boldsymbol{\eta}\|_2^2 + 2\|\mathbf{Ax} - \mathbf{A}\bar{\mathbf{x}}\|_2 \|P(\mathbf{A}_{\mathbf{S}\cup\mathbf{S}^*})\boldsymbol{\eta}\|_2$$
$$\leq \|\mathbf{Ax} - \mathbf{y}\|_2^2 - \|\boldsymbol{\eta}\|_2^2 + 2\|\mathbf{Ax} - \mathbf{A}\bar{\mathbf{x}}\|_2 \, \sigma\sqrt{s+s^*}\sqrt{\log\frac{d}{\delta}}$$

(w.p. at least $1 - 2\delta$)

$$\leq \|\mathbf{A}\bar{\mathbf{x}} - \mathbf{y}\|_2^2 + 40\frac{\rho_1^+}{\rho_{s+s^*}^+}\sigma^2 s \log\frac{d}{\delta} - \|\boldsymbol{\eta}\|_2^2 + 4\|\mathbf{Ax} - \mathbf{A}\bar{\mathbf{x}}\|_2 \, \sigma\sqrt{s}\sqrt{\log\frac{d}{\delta}}$$

(w.p. at least $1 - 2\delta$, using (3.10))

$$\leq 40\sigma^2 s \log\frac{d}{\delta} + 4\|\mathbf{Ax} - \mathbf{A}\bar{\mathbf{x}}\|_2 \, \sigma\sqrt{s}\sqrt{\log\frac{d}{\delta}}$$

Let us define $G := \|\mathbf{Ax} - \mathbf{A}\bar{\mathbf{x}}\|_2$ and $\alpha := \sigma\sqrt{s \log d/\delta}$. Then, we have

$$G^2 - 4G\alpha \leq 40\alpha^2$$
$$\implies (G - 2\alpha)^2 \leq (4 + 40)\alpha^2$$
$$\implies G \leq (2 + \sqrt{44})\alpha$$

Replacing $s \leq C_1 \widetilde{\kappa}_{s+s^*} s^* \log 5\rho_{s+s^*}^+/\rho_{s+s^*}^-$, we get for some constant $C_2 \leq (2 + \sqrt{44})\, C_1 \leq 9C_1$,

$$\frac{\|\mathbf{Ax} - \mathbf{A}\bar{\mathbf{x}}\|_2^2}{n} \leq C_2 \frac{\sigma^2 s}{n} \log\frac{d}{\delta}$$
$$\frac{\|\mathbf{Ax} - \mathbf{A}\bar{\mathbf{x}}\|_2^2}{n} \leq C_2 \frac{\sigma^2 s^*}{n} \widetilde{\kappa}_{s+s^*} \log\kappa_{s+s^*} \log\frac{d}{\delta}$$

$\square$

*Proof of Corollary 3.3.1.* From [1, Lemma 6] it immediately implies that RSS and RSC at sparsity level $k$ hold with probability at least $1 - e^{-C_0 n}$ with

$$\rho_k^- = \frac{n}{2}\sigma_{\min}(\boldsymbol{\Sigma}) - C_1 k \log d$$
$$\text{and, } \rho_k^+ = 2n\sigma_{\max}(\boldsymbol{\Sigma}) + C_1 k \log d$$

where $C_0, C_1$ are universal constants. For $k = s + s^*$, and since $n > 4C_1 \frac{k \log d}{\sigma_{\min}(\boldsymbol{\Sigma})}$ we have

$$\rho_{s+s^*}^- \geq \frac{n\sigma_{\min}(\boldsymbol{\Sigma})}{4}$$
$$\text{and, } \rho_{s+s^*}^+(\boldsymbol{\Sigma}) \leq 2.25 n\sigma_{\max}(\boldsymbol{\Sigma})$$

Therefore it is enough to choose $s \geq 72\kappa(\boldsymbol{\Sigma}) \log(45\kappa(\boldsymbol{\Sigma})) s^*$ to apply Theorem 3.3. $\qquad \square$

*Proof of Corollary 3.3.2.* Using results from the proof of 3.3.1, we can specialize for the case when $\boldsymbol{\Sigma} = \mathbf{I}$. To invoke Theorem 3.1 we are required to see how and when does $\gamma$ concentrate as $n$ increases for a given $s$. From the definition of $\gamma$, we want to upper bound

$$
\begin{aligned}
\max_{\mathbf{S}} \left\| \mathbf{A}_{\mathbf{S}^* \setminus \mathbf{S}}^T \mathbf{A}_{\mathbf{S}} (\mathbf{A}_{\mathbf{S}}^T \mathbf{A}_{\mathbf{S}})^{-1} \right\|_\infty &= \max_{\mathbf{S}} \max_{i \in \mathbf{S}^* \setminus \mathbf{S}} \left\| \mathbf{A}_i^T \mathbf{A}_{\mathbf{S}} (\mathbf{A}_{\mathbf{S}}^T \mathbf{A}_{\mathbf{S}})^{-1} \right\|_1 \\
&\leq \sqrt{s} \max_{\mathbf{S}} \max_{i \in \mathbf{S}^* \setminus \mathbf{S}} \left\| \mathbf{A}_i^T \mathbf{A}_{\mathbf{S}} (\mathbf{A}_{\mathbf{S}}^T \mathbf{A}_{\mathbf{S}})^{-1} \right\|_2 \\
&\leq \sqrt{s} \max_{\mathbf{S}} \max_{i \in \mathbf{S}^* \setminus \mathbf{S}} \left\| \mathbf{A}_{\mathbf{S}}^T \mathbf{A}_i \right\|_2 \left\| (\mathbf{A}_{\mathbf{S}}^T \mathbf{A}_{\mathbf{S}})^{-1} \right\|_2 \\
&\leq s \max_{\mathbf{S}} \max_{i \in \mathbf{S}^* \setminus \mathbf{S}} \left\| \mathbf{A}_{\mathbf{S}}^T \mathbf{A}_i \right\|_\infty \left\| (\mathbf{A}_{\mathbf{S}}^T \mathbf{A}_{\mathbf{S}})^{-1} \right\|_2 \\
&\leq s \max_{\mathbf{S}} \max_{i \in \mathbf{S}^* \setminus \mathbf{S}} \left\| \mathbf{A}_{(\mathbf{S}^* \setminus \mathbf{S})^c}^T \mathbf{A}_i \right\|_\infty \left\| (\mathbf{A}_{\mathbf{S}}^T \mathbf{A}_{\mathbf{S}})^{-1} \right\|_2 \\
&\leq s \max_{\mathbf{S}} \max_{i \in \mathbf{S}^* \setminus \mathbf{S}} \left\| \mathbf{A}_{(\mathbf{S}^* \setminus \mathbf{S})^c}^T \mathbf{A}_i \right\|_\infty \cdot \max_{\mathbf{S}} \left\| (\mathbf{A}_{\mathbf{S}}^T \mathbf{A}_{\mathbf{S}})^{-1} \right\|_2 \\
&\leq s \max_{\mathbf{S}} \max_{i \in \mathbf{S}^* \setminus \mathbf{S}} \left\| \mathbf{A}_{(\mathbf{S}^* \setminus \mathbf{S})^c}^T \mathbf{A}_i \right\|_\infty \frac{1}{\rho_s^-} \\
&\leq s \max_{\mathbf{S}} \max_{i \in \mathbf{S}^* \setminus \mathbf{S}} \left\| \mathbf{A}_{(\mathbf{S}^* \setminus \mathbf{S})^c}^T \mathbf{A}_i \right\|_\infty \frac{1}{n/4} \\
&\quad \text{(w.p. at least } 1 - e^{-C_0 n} \text{ if } n \geq 4Cs \log d) \\
&\leq \frac{4s}{n} \max_{\mathbf{S}} \max_{i \in \mathbf{S}^* \setminus \mathbf{S}} \left\| \mathbf{A}_{(\mathbf{S}^* \setminus \mathbf{S})^c}^T \mathbf{A}_i \right\|_\infty \qquad (B.33)
\end{aligned}
$$

To bound the quantity in (B.33) we can use the vanilla Chernoff bound. Note that it is sufficient bounding the maximum of $\binom{d}{2}$ inner products of pair of $d$-length standard Gaussian vectors. For any two independent standard Gaussian vectors $\mathbf{u}$ and $\mathbf{v}$ in $\mathbb{R}^n$, we will have

$$
\begin{aligned}
P\{\langle \mathbf{u}, \mathbf{v} \rangle \geq \epsilon\} = P\{\exp\{t\langle \mathbf{u}, \mathbf{v}\rangle\} \geq e^{t\epsilon}\} \\
\leq \inf_{t>0} e^{-t\epsilon} \mathbb{E}\left[e^{tZ}\right]^n \qquad (Z \text{ is a product of two standard normal random variables}) \\
\leq \inf_{t>0} \exp\left\{t^2 \frac{n}{2} - t\epsilon\right\} \\
= \exp\left\{-\frac{\epsilon^2}{2n}\right\} \qquad (B.34)
\end{aligned}
$$

We need to find $\epsilon$, so for a moment let us bound the RHS in (B.34) with some appropriate probability $\delta'$ which we will define later. This gives us

$$\exp\left\{-\frac{\epsilon^2}{2n}\right\} \leq \delta'$$

$$\implies \epsilon \geq \sqrt{2n\log\frac{1}{\delta'}}$$

$$P\left\{\langle \mathbf{u}, \mathbf{v}\rangle \geq \sqrt{2n\log\frac{1}{\delta'}}\right\} \leq \delta'$$

$$\text{or, } P\left\{|\langle \mathbf{u}, \mathbf{v}\rangle| \geq \sqrt{2n\log\frac{1}{\delta'}}\right\} \leq 2\delta'$$

Because we have $\binom{d}{2}$ such unique inner products, we set $\delta' = \delta/\binom{d}{2}$ and use a union bound to get

$$P\left\{\max_{\mathbf{S}}\max_{i\in\mathbf{S}^*\backslash\mathbf{S}}\left\|\mathbf{A}_{(\mathbf{S}^*\backslash\mathbf{S})^c}^T\mathbf{A}_i\right\|_\infty \leq 2\sqrt{n\log\frac{d}{\delta}}\right\} \geq P\left\{\max_{\mathbf{S}}\max_{i\in\mathbf{S}^*\backslash\mathbf{S}}\left\|\mathbf{A}_{(\mathbf{S}^*\backslash\mathbf{S})^c}^T\mathbf{A}_i\right\|_\infty \leq \sqrt{2n\log\left(\binom{d}{2}/\delta\right)}\right\}$$

$$\geq 1 - 2\delta \tag{B.35}$$

Using (B.35) in (B.33) we finally get

$$\max_{\mathbf{S}}\left\|\mathbf{A}_{\mathbf{S}^*\backslash\mathbf{S}}^T\mathbf{A}_{\mathbf{S}}(\mathbf{A}_{\mathbf{S}}^T\mathbf{A}_{\mathbf{S}})^{-1}\right\|_\infty \leq 8s\sqrt{\frac{\log(d/\delta)}{n}} \qquad \text{(with probabily at least } 1 - e^{-C_0 n} - 2\delta) \tag{B.36}$$

And if further $n \geq 64s^2\log\frac{d}{\delta}$, we have $\gamma \leq 1$ with probability at least $1 - e^{-C_0 n} - 2\delta$. Ones we have $\gamma$ bounded by some constant, one can easily check that the requirement on $|\bar{x}_{\min}|$ becomes

$$|\bar{x}_{\min}| \geq 23\sigma\sqrt{\log\frac{d}{\delta}} \tag{B.37}$$

Given this condition, using Theorem 3.1 we have support recovery as well as an infinity norm bound on the parameter space

$$\|\hat{\mathbf{x}}_s - \bar{\mathbf{x}}\|_\infty \leq 2\sigma\sqrt{\frac{2\log(s/\delta)}{n}} \tag{B.38}$$

which completes the proof. $\qquad\square$

## C  Proofs of results in Section 4

*Proof of Lemma 4.1.* Consider $\mathbf{v}$ to be a sparse vector such that $\|\mathbf{v}\|_0 = s$ and $\|\mathbf{v}\|_2 = 1$. Let $\mathbf{S} := \operatorname{supp}(\mathbf{v})$. For the rest of the proof, define $\mathbf{A} := \mathbf{M}^{(\epsilon)}$. This induces a sub-matrix $\mathbf{A}_{\mathbf{S}}$ which is the matrix of columns of $\mathbf{A}$ corresponding to the support of $\mathbf{v}$. Let us further split $\mathbf{S} = (\mathbf{S}\cap\mathbf{S}^*)\cup(\mathbf{S}\backslash\mathbf{S}^*)$ where $\mathbf{S}^* = [s^*]$. Consider a vector $\bar{\mathbf{x}}$ such that $\bar{x}_i = 1/s^*$ when $i \in [s^*]$ and 0 otherwise. Then, $\mathbf{A}_{\mathbf{S}\backslash\mathbf{S}^*} = \sqrt{1-\epsilon}\,\mathbf{A}_{\mathbf{S}^*}\bar{\mathbf{x}}\mathbf{1}_{|\mathbf{S}\backslash\mathbf{S}^*|}^T + \sqrt{\epsilon}\,\mathbf{G}$, where $\mathbf{G}$ is the $(n \times |\mathbf{S}\backslash\mathbf{S}^*|)$ matrix containing column vectors $\mathbf{g}_i$ for $i \in \mathbf{S}\backslash\mathbf{S}^*$.

$$\|\mathbf{A_S v}\|_2^2 = \left\| \sqrt{1-\epsilon} \mathbf{A_{S^*}} \bar{\mathbf{x}} \mathbf{1}^T \mathbf{v_{S\setminus S^*}} + \sqrt{\epsilon} \mathbf{G v_{S\setminus S^*}} + \mathbf{A_{S\cap S^*}} \mathbf{v_{S\cap S^*}} \right\|_2^2$$

$$= \left\| \sum_{i\in \mathbf{S^*}} \sum_{j\in \mathbf{S\setminus S^*}} (v_j \sqrt{1-\epsilon}\, \bar{\mathbf{x}}_i) \mathbf{A}_i + \sqrt{\epsilon} \mathbf{G v_{S\setminus S^*}} + \mathbf{A_{S\cap S^*}} \mathbf{v_{S\cap S^*}} \right\|_2^2$$

$$= \left\| \sum_{i\in \mathbf{S^*\setminus S}} \sum_{j\in \mathbf{S\setminus S^*}} (v_j \sqrt{1-\epsilon}\, \bar{\mathbf{x}}_i) \mathbf{A}_i + \sqrt{\epsilon} \mathbf{G v_{S\setminus S^*}} + \sum_{i\in \mathbf{S\cap S^*}} \left[ \left( \sum_{j\in \mathbf{S\setminus S^*}} v_j \sqrt{1-\epsilon}\, \bar{\mathbf{x}}_i \right) + v_i \right] \mathbf{A}_i \right\|_2^2$$

$$= n(1-\epsilon) \sum_{i\in \mathbf{S^*\setminus S}} \left( \sum_{j\in \mathbf{S\setminus S^*}} v_j \bar{\mathbf{x}}_i \right)^2 + \epsilon \|\mathbf{G v_{S\setminus S^*}}\|_2^2 + n \sum_{i\in \mathbf{S\cap S^*}} \left[ \left( \sum_{j\in \mathbf{S\setminus S^*}} v_j \sqrt{1-\epsilon}\, \bar{\mathbf{x}}_i \right) + v_i \right]^2 \tag{C.1}$$

Define $p := \sum_{j\in \mathbf{S\setminus S^*}} v_j, q := \sum_{j\in \mathbf{S\setminus S^*}} v_j^2, r := \sum_{i\in \mathbf{S\cap S^*}} v_i, t := \sum_{i\in \mathbf{S\cap S^*}} v_i^2 = 1-q$.
Since $\bar{\mathbf{x}}_i = 1/\sqrt{s^*}$,

$$\|\mathbf{A_S v}\|_2^2 = n(1-\epsilon)p^2 \frac{|\mathbf{S^*\setminus S}|}{s^*} + n\epsilon q + n \sum_{i\in \mathbf{S^*\cap S}} \left( p\sqrt{\frac{1-\epsilon}{s^*}} + v_i \right)^2$$

$$= n(1-\epsilon)p^2 + n\epsilon q + 2npr\sqrt{\frac{1-\epsilon}{s^*}} + nt. \tag{C.2}$$

We need to lower bound this quantity to get an estimate for $\rho_s^-$. Then,

$$\|\mathbf{A_S v}\|_2^2 = n(1-\epsilon)p^2 + n\epsilon q + 2npr\sqrt{\frac{1-\epsilon}{s^*}} + nt$$

$$\geq n(1-\epsilon)p^2 + n\epsilon q - 2np\sqrt{t}\sqrt{1-\epsilon} + nt$$

$$= n(p\sqrt{1-\epsilon} - \sqrt{t})^2 + n\epsilon q.$$

Suppose $p\sqrt{1-\epsilon} < \sqrt{t/2}$, then the bound becomes $n(t+q)/4 = n/4$. Else, $\sqrt{1-\epsilon}\sqrt{sq} \geq p\sqrt{1-\epsilon} \geq \sqrt{t/2}$. Setting $t = 1-q$, we get:

$$2(1-\epsilon)sq \geq 1-q$$

$$\implies q \geq \frac{1}{1+2(1-\epsilon)s}$$

In any case, we get,

$$\rho_s^- = \min_{\mathbf{S}\subseteq[d], |\mathbf{S}|=s} \min_{\|\mathbf{v}\|_2^2=1} \|\mathbf{A_S v}\|_2^2 \geq \min\left\{ \frac{n}{4}, \frac{n}{4(1+2(1-\epsilon)s)} \right\} = \frac{n}{4(1+2(1-\epsilon)s)} \tag{C.3}$$

Therefore $\widetilde{\kappa}_s \leq 4(1+2(1-\epsilon)s)$.
Now, we wish to lower bound $\left\| \mathbf{A}_{\mathbf{S^*\setminus S}}^T \mathbf{A_S} \left( \mathbf{A_S^T A_S} \right)^{-1} \right\|_\infty$ even when some correct elements might get recovered. This is essentially the max $\ell_1$ norm of the rows of the matrix $\mathbf{A}_{\mathbf{S^*\setminus S}}^T \mathbf{A_S} \left( \mathbf{A_S^T A_S} \right)^{-1}$, i.e.,

$$\left\| \mathbf{A}_{\mathbf{S^*\setminus S}}^T \mathbf{A_S} \left( \mathbf{A_S^T A_S} \right)^{-1} \right\|_\infty = \max_{i\in \mathbf{S^*\setminus S}} \left\| \left( \mathbf{A_S^T A_S} \right)^{-1} \mathbf{A_S^T A}_i \right\|_1 \tag{C.4}$$

$$\mathbf{A_S^T A}_i = n \begin{bmatrix} \mathbf{0}_{|\mathbf{S\cap S^*}|} \\ \mathbf{1}_{|\mathbf{S\setminus S^*}|} \sqrt{\frac{1-\epsilon}{s^*}} \end{bmatrix} \tag{C.5}$$

From (C.5) we see that we only requires the sub-matrix of $\left(\mathbf{A}_{\mathbf{S}}^T \mathbf{A}_{\mathbf{S}}\right)^{-1}$ with columns corresponding to the index set $\mathbf{S} \backslash \mathbf{S}^*$.

Let $p := |\mathbf{S} \cap \mathbf{S}^*|$ and $q := \mathbf{S} \backslash \mathbf{S}^*$. From block matrix inversion we have that

$$
\left[\left(\mathbf{A}_{\mathbf{S}}^T \mathbf{A}_{\mathbf{S}}\right)^{-1}\right]_{\mathbf{S} \backslash \mathbf{S}^*} = \frac{1}{n} \begin{bmatrix} \mathbf{I}_p & \sqrt{\frac{1-\epsilon}{s^*}} \mathbf{1}_p \mathbf{1}_q^T \\ \sqrt{\frac{1-\epsilon}{s^*}} \mathbf{1}_q \mathbf{1}_p^T & (1-\epsilon)\mathbf{1}_q \mathbf{1}_q^T + \epsilon \mathbf{I}_q \end{bmatrix}_{\mathbf{S} \backslash \mathbf{S}^*}^{-1}
$$

$$
= \frac{1}{n} \begin{bmatrix} -\sqrt{\frac{1-\epsilon}{s^*}} \mathbf{1}_p \mathbf{1}_q^T \left[(1-\epsilon)\mathbf{1}_q \mathbf{1}_q + \epsilon \mathbf{I}_q - \frac{1-\epsilon}{s^*} p \mathbf{1}_q \mathbf{1}_q^T\right]^{-1} \\ \left[(1-\epsilon)\mathbf{1}_q \mathbf{1}_q^T + \epsilon \mathbf{I}_q - \frac{1-\epsilon}{s^*} p \mathbf{1}_q \mathbf{1}_q^T\right]^{-1} \end{bmatrix} \quad \text{(C.6)}
$$

Let us separately compute $\left[(1-\epsilon)\mathbf{1}_q \mathbf{1}_q^T + \epsilon \mathbf{I}_q - \frac{1-\epsilon}{s^*} p \mathbf{1}_q \mathbf{1}_q^T\right]^{-1}$ first

$$
\left[(1-\epsilon)\mathbf{1}_q \mathbf{1}_q^T + \epsilon \mathbf{I}_q - \frac{1-\epsilon}{s^*} p \mathbf{1}_q \mathbf{1}_q^T\right]^{-1} = \left[\epsilon \mathbf{I}_q + (1-\epsilon)\left(1 - \frac{p}{s^*}\right)\mathbf{1}_q \mathbf{1}_q\right]^{-1}
$$

$$
= \frac{1}{\epsilon}\mathbf{I}_q - \frac{\frac{1}{\epsilon^2}(1-\epsilon)\left(1 - \frac{p}{s^*}\right)}{1 + (1-\epsilon)\left(1 - \frac{p}{s^*}\right)\frac{q}{\epsilon}} \mathbf{1}_q \mathbf{1}_q^T
$$

$$
= \frac{1}{\epsilon}\left[\mathbf{I}_q - \frac{(1-\epsilon)\left(1 - \frac{p}{s^*}\right)}{\epsilon + (1-\epsilon)\left(1 - \frac{p}{s^*}\right)q} \mathbf{1}_q \mathbf{1}_q^T\right] \quad \text{(C.7)}
$$

Using (C.7) we have

$$
-\sqrt{\frac{1-\epsilon}{s^*}} \mathbf{1}_p \mathbf{1}_q^T \left[(1-\epsilon)\mathbf{1}_q \mathbf{1}_q + \epsilon \mathbf{I}_q - \frac{1-\epsilon}{s^*} p \mathbf{1}_q \mathbf{1}_q^T\right]^{-1} \quad \text{(C.8)}
$$

$$
= -\sqrt{\frac{1-\epsilon}{s^*}} \mathbf{1}_p \mathbf{1}_q^T \frac{1}{\epsilon}\left[\mathbf{I}_q - \frac{(1-\epsilon)\left(1 - \frac{p}{s^*}\right)}{\epsilon + (1-\epsilon)\left(1 - \frac{p}{s^*}\right)q} \mathbf{1}_q \mathbf{1}_q^T\right]
$$

$$
= -\sqrt{\frac{1-\epsilon}{s^*}} \mathbf{1}_p \frac{1}{\epsilon}\left[\mathbf{1}_q^T - \frac{(1-\epsilon)\left(1 - \frac{p}{s^*}\right)q}{\epsilon + (1-\epsilon)\left(1 - \frac{p}{s^*}\right)q} \mathbf{1}_q^T\right]
$$

$$
= -\sqrt{\frac{1-\epsilon}{s^*}} \left[\frac{1}{\epsilon + (1-\epsilon)\left(1 - \frac{p}{s^*}\right)q}\right] \mathbf{1}_p \mathbf{1}_q^T \quad \text{(C.9)}
$$

Using (C.7) and (C.9) we have

$$
\left(\mathbf{A}_{\mathbf{S}}^T \mathbf{A}_{\mathbf{S}}\right)^{-1} \mathbf{A}_{\mathbf{S}}^T \mathbf{A}_i = \sqrt{\frac{1-\epsilon}{s^*}} \begin{bmatrix} -\sqrt{\frac{1-\epsilon}{s^*}}\left[\frac{1}{\epsilon + (1-\epsilon)\left(1 - \frac{p}{s^*}\right)q}\right] \mathbf{1}_p \mathbf{1}_q^T \\ \frac{1}{\epsilon}\left[\mathbf{I}_q - \frac{(1-\epsilon)\left(1 - \frac{p}{s^*}\right)}{\epsilon + (1-\epsilon)\left(1 - \frac{p}{s^*}\right)q} \mathbf{1}_q \mathbf{1}_q^T\right] \end{bmatrix} \mathbf{1}_q
$$

$$
= \sqrt{\frac{1-\epsilon}{s^*}} \begin{bmatrix} -\sqrt{\frac{1-\epsilon}{s^*}}\left[\frac{q}{\epsilon + (1-\epsilon)\left(1 - \frac{p}{s^*}\right)q}\right] \mathbf{1}_p \\ \frac{1}{\epsilon}\left[\frac{\epsilon}{\epsilon + (1-\epsilon)\left(1 - \frac{p}{s^*}\right)q}\right] \mathbf{1}_q \end{bmatrix}
$$

$$
= -\sqrt{\frac{1-\epsilon}{s^*}} \frac{1}{\epsilon + (1-\epsilon)\left(1 - \frac{p}{s^*}\right)q} \begin{bmatrix} -q\sqrt{\frac{1-\epsilon}{s^*}} \mathbf{1}_p \\ \mathbf{1}_q \end{bmatrix}
$$

$$
\implies \left\|\left(\mathbf{A}_{\mathbf{S}}^T \mathbf{A}_{\mathbf{S}}\right)^{-1} \mathbf{A}_{\mathbf{S}}^T \mathbf{A}_i\right\|_1 = \sqrt{\frac{1-\epsilon}{s^*}} \frac{q}{\epsilon + (1-\epsilon)\left(1 - \frac{p}{s^*}\right)q}\left[p\sqrt{\frac{1-\epsilon}{s^*}} + 1\right] \quad \text{(C.10)}
$$

Therefore $\left\|\mathbf{A}_{\mathbf{S}^* \backslash \mathbf{S}}^T \mathbf{A}_{\mathbf{S}}\left(\mathbf{A}_{\mathbf{S}}^T \mathbf{A}_{\mathbf{S}}\right)^{-1}\right\|_\infty = \sqrt{\frac{1-\epsilon}{s^*}} \frac{q}{\epsilon + (1-\epsilon)\left(1 - \frac{p}{s^*}\right)q}\left[p\sqrt{\frac{1-\epsilon}{s^*}} + 1\right]$. $\qquad \square$

*Proof of Theorem 4.2.* We shall construct a matrix $\mathbf{A} \in \mathbb{R}^{n \times d}$ where $n \geq d$.

Having a closer look at Algorithm 1 for sparse linear regression , we can see that $\nabla Q(\mathbf{x}_k) = \frac{2}{n}\mathbf{A}^T(\mathbf{A}\mathbf{x}_k - \mathbf{y})$. The residual vector is now $\mathbf{r}_k := \mathbf{y} - \mathbf{A}\mathbf{x}_k$ and hence the selected index is $j$ is

$\arg\max_{i\in\mathbf{S}_k^c}|\mathbf{A}_i^T\mathbf{r}_k|$. Therefore with proper construction of the matrix $\mathbf{A}$, we can force the new selected index to be in $(\mathbf{S}^*)^c$.

We first fix the vector to be estimated, $\bar{\mathbf{x}}$, as $\bar{x}_i = {}^1/\sqrt{s^*}\ \forall\ i\in[s^*]$ and 0 otherwise. Thus, $\mathbf{S}^* = [s^*]$ and we take $s^* \geq 2$. We define $\mathbf{A} := \mathbf{M}^{(\epsilon)}$ and choose a constant $\epsilon$ in the range $[1 - {}^3/2s^*, 1 - {}^1/s^*)$. We assume that OMP fails to recover any support by the $(k-1)^{th}$ iterate, i.e., $\mathbf{S}_{k-1}\cap\mathbf{S} = \phi$. At the $k^{th}$ OMP iteration, for $i\notin(\mathbf{S}_{k-1}\cup\mathbf{S}^*)$,

$$\begin{aligned}
|\mathbf{A}_i^T\mathbf{r}_{k-1}| &= |\sqrt{1-\epsilon}\,\mathbf{y}^T\mathbf{r}_{k-1} + \sqrt{\epsilon}\mathbf{g}_i^T\mathbf{r}_{k-1}|\\
&= \sqrt{1-\epsilon}|\mathbf{y}^T\mathbf{r}_{k-1}|\\
&= \sqrt{1-\epsilon}\sum_{j\in\mathbf{S}^*}\bar{x}_j|\mathbf{r}_{k-1}^T\mathbf{A}_j|
\end{aligned}\tag{C.11}$$

Now let us consider the term $\mathbf{r}_{k-1}^T\mathbf{A}_j$ for all $j\in\mathbf{S}^*$.

$$\begin{aligned}
\mathbf{r}_{k-1}^T\mathbf{A}_j &= \mathbf{y}^T(\mathbf{I} - \mathbf{A}_{\mathbf{S}_{k-1}}(\mathbf{A}_{\mathbf{S}_{k-1}}^T\mathbf{A}_{\mathbf{S}_{k-1}})^{-1}\mathbf{A}_{\mathbf{S}_{k-1}}^T)\mathbf{A}_j\\
&\overset{\xi_1}{=} \mathbf{y}^T\mathbf{A}_j - \left[\mathbf{y}^T\mathbf{A}_{\mathbf{S}_{k-1}}(\mathbf{A}_{\mathbf{S}_{k-1}}^T\mathbf{A}_{\mathbf{S}_{k-1}})^{-1}\mathbf{1}\right](\sqrt{1-\epsilon}\,\mathbf{y}^T\mathbf{A}_j)\\
&= (\mathbf{A}_{\mathbf{S}^*}\bar{\mathbf{x}})^T\mathbf{A}_j - \left[\mathbf{y}^T\mathbf{A}_{\mathbf{S}_{k-1}}(\mathbf{A}_{\mathbf{S}_{k-1}}^T\mathbf{A}_{\mathbf{S}_{k-1}})^{-1}\mathbf{1}\right](\sqrt{1-\epsilon}\,(\mathbf{A}_{\mathbf{S}^*}\bar{\mathbf{x}})^T\mathbf{A}_j)\\
&= n\bar{x}_j - \left[\mathbf{y}^T\mathbf{A}_{\mathbf{S}_{k-1}}(\mathbf{A}_{\mathbf{S}_{k-1}}^T\mathbf{A}_{\mathbf{S}_{k-1}})^{-1}\mathbf{1}\right](\sqrt{1-\epsilon}\,(n\bar{x}_j))
\end{aligned}$$

Here, $\xi_1$ holds because $\mathbf{g}_l^T\mathbf{A}_j = 0, \forall\,l\in\mathbf{S}_{k-1}, j\in\mathbf{S}^*$. Now, because $\bar{x}_j = {}^1/\sqrt{s^*}$ for all index $j\in\mathbf{S}^*$, the above quantity is the same irrespective of what $j$ is. Thus, we can write (C.11) as

$$\begin{aligned}
|\mathbf{A}_i^T\mathbf{r}_{k-1}| &= \sqrt{1-\epsilon}\sum_{j\in\mathbf{S}^*}x_j^*|\mathbf{r}_{k-1}^T\mathbf{A}_j|\\
&= \sqrt{1-\epsilon}\sqrt{s^*}\max_{j\in\mathbf{S}^*}|\mathbf{r}_{k-1}^T\mathbf{A}_j|
\end{aligned}\tag{C.12}$$

Therefore for $\epsilon$ such that $\sqrt{1-\epsilon}\sqrt{s^*} > 1$, or when $\epsilon < 1 - \frac{1}{s^*}$, then $|\mathbf{A}_i^T\mathbf{r}_{k-1}| > \max_{j\in\mathbf{S}^*}|\mathbf{A}_j^T\mathbf{r}_{k-1}|$

which would imply that the algorithm in the $k^{th}$ iteration picks an incorrect index that is not contained in $\mathbf{S}^*$.

Using Lemma 4.1, setting $\epsilon = 1 - {}^3/2s^*$, we get

$$\widetilde{\kappa}_s(\mathbf{A}) \leq \frac{16s}{s^*} \text{ and } \gamma \leq \sqrt{\frac{2}{3}}\tag{C.13}$$

$\square$

*Proof of Theorem 4.3.* When the model is noisy, we consider a similar matrix $\mathbf{A} := \mathbf{M}^{(\epsilon)}$ discussed in 4.1 where we set $\epsilon = 1 - {}^4/s^*$. Further, we are concerned about the asymptotic failure of OMP in the presence of a large number of samples. So we set $n \geq 4\sigma^2s^2\log{}^d/\delta$. Suppose that OMP has not recovered any support until $k$ iterations. We show that even in the next (ie, $(k+1)^{th}$) iteration, an incorrect support is picked. The criteria for OMP to not select an index in $\mathbf{S}^*$ is

$$\max_{j\notin(\mathbf{S}^*\cup\mathbf{S}_k)}|\mathbf{A}_j^T\mathbf{r}_k| \geq \max_{i\in\mathbf{S}^*}|\mathbf{A}_j^T\mathbf{r}_k|\tag{C.14}$$

We will show that the above inequality holds with good probability. We denote $\mathbf{1}$ as the vector with all its elements as 1. Denote the projection matrix corresponding to a matrix $\mathbf{A}_{\mathbf{S}_k}$ as $P(\mathbf{A}_{\mathbf{S}_k}) := \mathbf{A}_{\mathbf{S}_k}(\mathbf{A}_{\mathbf{S}_k}^T\mathbf{A}_{\mathbf{S}_k})^{-1}\mathbf{A}_{\mathbf{S}_k}^T$. Note that $\mathbf{r}_k = (\mathbf{I} - P(\mathbf{A}_{\mathbf{S}_k}))(\mathbf{A}_{\mathbf{S}^*}\bar{\mathbf{x}} + \boldsymbol{\eta})$.

$$\begin{aligned}
|\mathbf{A}_j^T\mathbf{r}_k| &= |\mathbf{A}_j^T(\mathbf{I} - P(\mathbf{A}_{\mathbf{S}_k}))(\mathbf{A}_{\mathbf{S}^*}\bar{\mathbf{x}} + \boldsymbol{\eta})|\\
&= |\mathbf{A}_j^T\mathbf{A}_{\mathbf{S}^*}\bar{\mathbf{x}} - \mathbf{A}_j^TP(\mathbf{A}_{\mathbf{S}_k})\mathbf{A}_{\mathbf{S}^*}\bar{\mathbf{x}} + \mathbf{A}_j^T\boldsymbol{\eta} - \mathbf{A}_j^TP(\mathbf{A}_{\mathbf{S}_k})\boldsymbol{\eta}|
\end{aligned}\tag{C.15}$$

We can analyze these terms separately as below

$$\mathbf{A}_j^T \mathbf{A}_{\mathbf{S}^*} \bar{\mathbf{x}} = (\sqrt{1-\epsilon} \mathbf{A}_{\mathbf{S}^*} \bar{\mathbf{x}} + \sqrt{\epsilon} \mathbf{g}_j)^T \mathbf{A}_{\mathbf{S}^*} \bar{\mathbf{x}}$$

$$= n\sqrt{1-\epsilon} \tag{C.16}$$

$$\mathbf{A}_j^T P(\mathbf{A}_{\mathbf{S}_k}) \mathbf{A}_{\mathbf{S}^*} \bar{\mathbf{x}} = \mathbf{A}_j^T \mathbf{A}_{\mathbf{S}_k} (\mathbf{A}_{\mathbf{S}_k}^T \mathbf{A}_{\mathbf{S}_k})^{-1} \mathbf{A}_{\mathbf{S}_k}^T \mathbf{A}_{\mathbf{S}^*} \bar{\mathbf{x}}$$

$$= n(1-\epsilon) \mathbf{1}^T (\mathbf{A}_{\mathbf{S}_k}^T \mathbf{A}_{\mathbf{S}_k})^{-1} \mathbf{A}_{\mathbf{S}_k}^T \mathbf{A}_{\mathbf{S}^*} \bar{\mathbf{x}}$$

$$= \frac{n(1-\epsilon)}{n(1+(k-1)(1-\epsilon))} \mathbf{1}^T \mathbf{A}_{\mathbf{S}_k}^T \mathbf{A}_{\mathbf{S}^*} \bar{\mathbf{x}} \quad (\because \mathbf{1} \text{ is an eigenvector of } \mathbf{A}_{\mathbf{S}_k}^T \mathbf{A}_{\mathbf{S}_k})$$

$$= \frac{1-\epsilon}{1+(k-1)(1-\epsilon)} \mathbf{1}^T \mathbf{1} n \sqrt{1-\epsilon}$$

$$= \frac{nk(1-\epsilon)\sqrt{1-\epsilon}}{1+(k-1)(1-\epsilon)} \tag{C.17}$$

$$\mathbf{A}_j^T P(\mathbf{A}_{\mathbf{S}_k}) \boldsymbol{\eta} = \frac{1-\epsilon}{1+(k-1)(1-\epsilon)} \mathbf{1}^T \mathbf{A}_{\mathbf{S}_k}^T \boldsymbol{\eta}$$

$$= \frac{1-\epsilon}{1+(k-1)(1-\epsilon)} \sum_{l \in \mathbf{S}_k} \mathbf{A}_l^T \boldsymbol{\eta} \tag{C.18}$$

Similarly we can analyze $\left| \mathbf{A}_i^T \mathbf{r}_k \right|$

$$\left| \mathbf{A}_i^T \mathbf{r}_k \right| = \left| \mathbf{A}_i^T (\mathbf{I} - P(\mathbf{A}_{\mathbf{S}_k})) (\mathbf{A}_{\mathbf{S}^*} \bar{\mathbf{x}} + \boldsymbol{\eta}) \right|$$

$$= \left| \mathbf{A}_i^T \mathbf{A}_{\mathbf{S}^*} \bar{\mathbf{x}} - \mathbf{A}_i^T P(\mathbf{A}_{\mathbf{S}_k}) \mathbf{A}_{\mathbf{S}^*} \bar{\mathbf{x}} + \mathbf{A}_i^T \boldsymbol{\eta} - \mathbf{A}_i^T P(\mathbf{A}_{\mathbf{S}_k}) \boldsymbol{\eta} \right| \tag{C.19}$$

We can analyze these terms separately as below

$$\mathbf{A}_i^T \mathbf{A}_{\mathbf{S}^*} \bar{\mathbf{x}} = \frac{n}{\sqrt{s^*}} \tag{C.20}$$

$$\mathbf{A}_i^T P(\mathbf{A}_{\mathbf{S}_k}) \mathbf{A}_{\mathbf{S}^*} \bar{\mathbf{x}} = \mathbf{A}_i^T \mathbf{A}_{\mathbf{S}_k} (\mathbf{A}_{\mathbf{S}_k}^T \mathbf{A}_{\mathbf{S}_k})^{-1} \mathbf{A}_{\mathbf{S}_k}^T \mathbf{A}_{\mathbf{S}^*} \bar{\mathbf{x}}$$

$$= \frac{n\sqrt{1-\epsilon}}{\sqrt{s^*}} \mathbf{1}^T (\mathbf{A}_{\mathbf{S}_k}^T \mathbf{A}_{\mathbf{S}_k})^{-1} \mathbf{A}_{\mathbf{S}_k}^T \mathbf{A}_{\mathbf{S}^*} \bar{\mathbf{x}}$$

$$= \frac{n\sqrt{1-\epsilon}}{\sqrt{s^*} n(1+(k-1)(1-\epsilon))} \mathbf{1}^T \mathbf{A}_{\mathbf{S}_k}^T \mathbf{A}_{\mathbf{S}^*} \bar{\mathbf{x}} \quad (\because \mathbf{1} \text{ is an eigenvector of } \mathbf{A}_{\mathbf{S}_k}^T \mathbf{A}_{\mathbf{S}_k})$$

$$= \frac{\sqrt{1-\epsilon}}{\sqrt{s^*}(1+(k-1)(1-\epsilon))} \mathbf{1}^T \mathbf{1} n \sqrt{1-\epsilon}$$

$$= \frac{nk(1-\epsilon)}{\sqrt{s^*}(1+(k-1)(1-\epsilon))} \tag{C.21}$$

$$\mathbf{A}_i^T P(\mathbf{A}_{\mathbf{S}_k}) \boldsymbol{\eta} = \frac{\sqrt{1-\epsilon}}{\sqrt{s^*}(1+(k-1)(1-\epsilon))} \mathbf{1}^T \mathbf{A}_{\mathbf{S}_k} \boldsymbol{\eta}$$

$$= \frac{\sqrt{1-\epsilon}}{\sqrt{s^*}(1+(k-1)(1-\epsilon))} \sum_{l \in \mathbf{S}_k} \mathbf{A}_l^T \boldsymbol{\eta} \tag{C.22}$$

Thus, for OMP to not recover a correct support at any iteration $k \leq s$, using (C.15) to (C.22), we have a required condition for all $i \in \mathbf{S}^* \setminus \mathbf{S}_{k-1}$, $j \in (\mathbf{S}^* \cup \mathbf{S}_{k-1})^c$.

$$\left| \mathbf{A}_i^T \mathbf{A}_{\mathbf{S}^*} \bar{\mathbf{x}} - \mathbf{A}_i^T P(\mathbf{A}_{\mathbf{S}_k}) \mathbf{A}_{\mathbf{S}^*} \bar{\mathbf{x}} + \mathbf{A}_i^T \boldsymbol{\eta} - \mathbf{A}_i^T P(\mathbf{A}_{\mathbf{S}_k}) \boldsymbol{\eta} \right|$$

$$\leq \left| \mathbf{A}_j^T \mathbf{A}_{\mathbf{S}^*} \bar{\mathbf{x}} - \mathbf{A}_j^T P(\mathbf{A}_{\mathbf{S}_k}) \mathbf{A}_{\mathbf{S}^*} \bar{\mathbf{x}} + \mathbf{A}_j^T \boldsymbol{\eta} - \mathbf{A}_j^T P(\mathbf{A}_{\mathbf{S}_k}) \boldsymbol{\eta} \right|$$

$$\text{or,} \quad \left| \frac{1}{\sqrt{s^*}} \frac{1}{1+(k-1)(1-\epsilon)} \left[ n\epsilon - \sqrt{1-\epsilon} \sum_{l \in \mathbf{S}_k} \mathbf{A}_l^T \boldsymbol{\eta} \right] + \mathbf{A}_i^T \boldsymbol{\eta} \right|$$

$$\leq \left| \frac{\sqrt{1-\epsilon}}{1+(k-1)(1-\epsilon)} \left[ n\epsilon - \sqrt{1-\epsilon} \sum_{l \in \mathbf{S}_k} \mathbf{A}_l^T \boldsymbol{\eta} \right] + \mathbf{A}_j^T \boldsymbol{\eta} \right| \tag{C.23}$$

Now, using triangle inequality on the LHS and RHS of C.23, we have a sufficient condition for failure:

$$\frac{1}{\sqrt{s^*}} \frac{1}{1+(k-1)(1-\epsilon)} \left| n\epsilon - \sqrt{1-\epsilon} \sum_{l \in \mathbf{S}_k} \mathbf{A}_l^T \boldsymbol{\eta} \right| + \left| \mathbf{A}_i^T \boldsymbol{\eta} \right|$$

$$\leq \frac{\sqrt{1-\epsilon}}{1+(k-1)(1-\epsilon)} \left| n\epsilon - \sqrt{1-\epsilon} \sum_{l \in \mathbf{S}_k} \mathbf{A}_l^T \boldsymbol{\eta} \right| - \left| \mathbf{A}_j^T \boldsymbol{\eta} \right|$$

Rearranging terms we get

$$\left| \mathbf{A}_i^T \boldsymbol{\eta} \right| + \left| \mathbf{A}_j^T \boldsymbol{\eta} \right| \leq \left( \sqrt{1-\epsilon} - \frac{1}{\sqrt{s^*}} \right) \frac{1}{1+(k-1)(1-\epsilon)} \left| n\epsilon - \sqrt{1-\epsilon} \sum_{l \in \mathbf{S}_k} \mathbf{A}_l^T \boldsymbol{\eta} \right| \quad \text{(C.24)}$$

Suppose $k \leq 1/9\sigma \sqrt{n/\log \frac{d}{\delta}}$. Because $\left\| \mathbf{A}^T \boldsymbol{\eta} \right\|_\infty \leq \sigma \sqrt{2n \log \frac{d}{\delta}}$ with probability at least $1 - \delta$,

$$n\epsilon \geq \sqrt{n}\epsilon\sqrt{n} \quad \text{(C.25)}$$

$$\geq 9k\sigma\epsilon \sqrt{n \log \frac{d}{\delta}}$$

$$\overset{\xi_1}{\geq} 9\sqrt{1-\epsilon}\, k\sigma \sqrt{2n \log \frac{d}{\delta}}$$

$$\geq 9\sqrt{1-\epsilon} \sum_{l \in \mathbf{S}_k} \mathbf{A}_l^T \boldsymbol{\eta}$$

Here, $\xi_1$ holds because $s^* \geq 8$. Then, a sufficient condition for (C.24) to be satisfied is,

$$2\sigma \sqrt{2n \log \frac{d}{\delta}} \leq \left( \sqrt{1-\epsilon} - \frac{1}{\sqrt{s^*}} \right) \frac{1}{1+(k-1)(1-\epsilon)} 8\sqrt{1-\epsilon} \left| \sum_{l \in \mathbf{S}_k} \mathbf{A}_l^T \boldsymbol{\eta} \right|$$

$$= \left( \sqrt{1-\epsilon} - \frac{1}{\sqrt{s^*}} \right) \frac{1}{1+(k-1)(1-\epsilon)} 8\sqrt{1-\epsilon} \sum_{l \in \mathbf{S}_k} \mathbf{A}_l^T \boldsymbol{\eta} \quad \text{(C.26)}$$

In the last equality ($\xi_1$), we remove absolute value signs by instantiating Lemma C.1 (with $\mathbf{T} \leftarrow$ k). The equality holds with probability larger than $1 - \delta$. We will now lower bound $\sum_{l \in \mathbf{S}_k} \mathbf{A}_l^T \boldsymbol{\eta}$. Consider the random variables $\mathbf{g}_j^T \boldsymbol{\eta}$ for $j \in [d] \setminus \mathbf{S}^*$ and $\mathbf{A}_i^T \boldsymbol{\eta}$ for $i \in \mathbf{S}^*$. Since $\mathbf{g}_j$'s and $\mathbf{A}_i$'s are orthogonal to each other, these can be looked at as independent Gaussian variables with variance $\sigma\sqrt{n}$. Now, define

$$X_i := \begin{cases} \frac{\mathbf{g}_i^T \boldsymbol{\eta}}{\sigma\sqrt{n}} & \text{for } i \in [d] \setminus \mathbf{S}^* \\ \frac{\mathbf{A}_i^T \boldsymbol{\eta}}{\sigma\sqrt{n}} & \text{for } i \in \mathbf{S}^* \end{cases} \quad \text{(C.27)}$$

Thus, $X_i$ are now independent standard Gaussian variables. We wish to show high probability bounds for the $k$-th order statistic of $X_i$'s. Suppose that the $k^{th}$ largest element of the set $\{X_1, X_2, \ldots, X_d\}$ be denoted as $\mathbf{X}^{(k)}$, then for all $t \geq 0$

$$P\left\{ X^{(k+1)} \leq t \right\} = \sum_{l=0}^{k} \binom{d}{l} \left[ P\left\{ X < t \right\} \right]^{d-l} \left[ P\left\{ X \geq t \right\} \right]^l$$

$$= \sum_{l=0}^{k} \binom{d}{l} \left[ 1 - \Phi(t) \right]^{d-l} \Phi(t)^l \quad \text{(C.28)}$$

The term on the right is the same as the probability that a binomial random variable $X$, with parameters $d$ and $\Phi(t)$ is less than $k$, and from [3] we know that it is sub-gamma on the left tail with variance factor less than $d\Phi(t)$ and scale factor $0$. Also a random variable $X$ is said to be sub-gamma on the right tail with variance $v$ and scale $c$ if

$$\log \mathbb{E} \left[ e^{\lambda(X - \mathbb{E}[X])} \right] \leq \frac{\lambda^2 v}{2(1 - c\lambda)} \qquad \forall \, \lambda > 0 \quad \text{(C.29)}$$

Also, $X$ is said to be sub-gamma on the left tail with variance factor $v$ and scale parameter $c$, if $-X$ is sub-gamma on the right tail with the same variance and scale parameters. Define $z := k - \mathbb{E}[X]$. From Chernoff bounds we have

$$
\begin{aligned}
P\left\{X^{(k+1)} \leq t\right\} &= P\left\{X \geq k\right\} \\
&= P\left\{X - \mathbb{E}[X] \geq k - \mathbb{E}[X]\right\} \\
&= P\left\{e^{\lambda(\mathbb{E}[X]-X)} \geq e^{\lambda z}\right\} \\
&\leq e^{-\lambda z}\mathbb{E}\left[e^{\lambda(\mathbb{E}[X]-X)}\right] \\
&\leq \exp\left\{-\lambda z + \frac{\lambda^2 v}{2}\right\} \\
&= \exp\left\{-\frac{z^2}{2v}\right\}
\end{aligned}
$$

Now, since $d \geq 4^{\frac{1}{\alpha}}$, $k \leq s \leq d^{1-\alpha} \leq \frac{d}{4} \leq \frac{d}{2} \leq d\Phi(t) = \mathbb{E}[X]$. Thus, $|z| \geq \left|d^{1-\alpha} - d\Phi(t)\right| \geq \left|d^{1-\alpha} - \frac{d}{2}\right|$. Further, we saw that the variance factor $v \leq d\Phi(t)$. Now, set $t = \sqrt{\log \frac{d}{k}}$. This gives,

$$
\begin{aligned}
P\left\{X^{(k+1)} \leq \sqrt{\log \frac{d}{k}}\right\} &\leq \exp\left\{-\frac{\left(d^{1-\alpha} - d\Phi\left(\left(\sqrt{\log \frac{d}{k}}\right)\right)\right)^2}{2d\Phi\left(\sqrt{\log \frac{d}{k}}\right)}\right\} \\
&\leq \exp\left\{-\frac{\left(d^{1-\alpha} - \frac{d}{2}\right)^2}{2d}\right\} \\
&\leq \exp\left\{-\frac{d\left(d^{-\alpha} - \frac{1}{2}\right)^2}{2}\right\} \\
&\leq \exp\left\{-\frac{d\left(\frac{1}{4} - \frac{1}{2}\right)}{2}\right\} \\
&\leq \exp\left\{-\frac{d}{32}\right\} \\
&\overset{\xi_1}{\leq} \delta
\end{aligned}
$$
(C.30)

Here, $\xi_1$ holds because $d \geq 32\log(1/\delta)$. Thus, for the top $k$ $\mathbf{g}_l$'s that maximize $\mathbf{g}_l^T \boldsymbol{\eta}$,

$$
P\left\{\mathbf{g}_l^T \boldsymbol{\eta} \geq \sigma\sqrt{n\log \frac{d}{k}}\right\} \geq 1 - \delta
$$
(C.31)

Now, with probability at least $1 - 2\delta$

$$
\begin{aligned}
\left|\mathbf{A}_l^T \boldsymbol{\eta}\right| &= \left|\sqrt{1-\epsilon}\,\mathbf{A}_S^* \bar{\mathbf{x}} + \sqrt{\epsilon}\mathbf{g}_l^T \boldsymbol{\eta}\right| \\
&\geq \sqrt{1-\epsilon}\,\sigma\sqrt{2n\log\left(\frac{d}{\delta}\right)} + \sqrt{\epsilon}\,\sigma\sqrt{n\log\left(\frac{d}{k}\right)} \\
&\geq \sqrt{1-\epsilon}\,\sigma\sqrt{2n\log\left(\frac{d}{k}\right)} + \sqrt{\epsilon}\,\sigma\sqrt{n\log\left(\frac{d}{k}\right)} \\
&\geq \sigma\sqrt{n\log\left(\frac{d}{k}\right)} \\
&\geq \sigma\sqrt{\alpha n\log d}
\end{aligned}
$$
(C.32)

Therefore $\left|\sum_{l\in\mathbf{S}_k}\mathbf{A}_l^T\boldsymbol{\eta}\right| \geq \sqrt{\alpha}\sigma k\sqrt{n\log d}$ with probability at least $1 - 2\delta$. Since, $\epsilon = 1 - 4/s^*$ and $s^* \geq 8$, then (C.26) reduces to,

$$k \geq \frac{(s^* - 4)\sqrt{\log d/\delta}}{2\sqrt{2\alpha\log d} - 4\sqrt{\log d/\delta}} \tag{C.33}$$

which is trivially satisfies because $s^* \geq 8$ and $\alpha < 1$. Therefore from the above argument, we have that since $k \leq \frac{1}{9\sigma}\sqrt{\frac{n}{\log d/\delta}}$, then with probability at least $1 - 2\delta$, an incorrect support is recovered at the $k^{th}$ OMP iterate.

Let us now have a look at the generalization error when none of the correct support is recovered, i.e., $\mathbf{S}_k \cap \mathbf{S}^* = \phi$.

$$\frac{1}{n}\left\|\mathbf{A}_{\mathbf{S}_k}\hat{\mathbf{x}}_k^{\text{OMP}} - \mathbf{A}_{\mathbf{S}^*}\bar{\mathbf{x}}\right\|_2^2 = \frac{1}{n}\left\|\mathbf{A}_{\mathbf{S}_k}(\mathbf{A}_{\mathbf{S}_k}^T\mathbf{A}_{\mathbf{S}_k})^{-1}\mathbf{A}_{\mathbf{S}_k}^T(\mathbf{A}_{\mathbf{S}^*}\bar{\mathbf{x}} + \boldsymbol{\eta}) - \mathbf{A}_{\mathbf{S}^*}\bar{\mathbf{x}}\right\|_2^2$$

$$= \frac{1}{n}\left\|\mathbf{A}_{\mathbf{S}_k}(\mathbf{A}_{\mathbf{S}_k}^T\mathbf{A}_{\mathbf{S}_k})^{-1}\mathbf{A}_{\mathbf{S}_k}^T\mathbf{A}_{\mathbf{S}^*}\bar{\mathbf{x}} + P(\mathbf{A}_{\mathbf{S}_k})\boldsymbol{\eta} - \mathbf{A}_{\mathbf{S}^*}\bar{\mathbf{x}}\right\|_2^2 \tag{C.34}$$

$$\mathbf{A}_{\mathbf{S}_k}(\mathbf{A}_{\mathbf{S}_k}^T\mathbf{A}_{\mathbf{S}_k})^{-1}\mathbf{A}_{\mathbf{S}_k}^T\mathbf{A}_{\mathbf{S}^*}\bar{\mathbf{x}} = \mathbf{A}_{\mathbf{S}_k}(\mathbf{A}_{\mathbf{S}_k}^T\mathbf{A}_{\mathbf{S}_k})^{-1}\mathbf{1}n\sqrt{1-\epsilon}$$

$$= \mathbf{A}_{\mathbf{S}_k}\mathbf{1}\frac{n\sqrt{1-\epsilon}}{n(1 + (k-1)(1-\epsilon))}$$

$$= \frac{1}{1 + (k-1)(1-\epsilon)}\left[k(1-\epsilon)\mathbf{A}_{\mathbf{S}^*}\bar{\mathbf{x}} + \sqrt{\epsilon(1-\epsilon)}\sum_{l\in\mathbf{S}_k}\mathbf{g}_i\right]$$

$$\implies \mathbf{A}_{\mathbf{S}^*}\bar{\mathbf{x}} - \mathbf{A}_{\mathbf{S}_k}(\mathbf{A}_{\mathbf{S}_k}^T\mathbf{A}_{\mathbf{S}_k})^{-1}\mathbf{A}_{\mathbf{S}_k}^T\mathbf{A}_{\mathbf{S}^*}\bar{\mathbf{x}} = \frac{1}{1 + (k-1)(1-\epsilon)}\left[\epsilon\mathbf{A}_{\mathbf{S}^*}\bar{\mathbf{x}} + \sqrt{\epsilon(1-\epsilon)}\sum_{l\in\mathbf{S}_k}\mathbf{g}_i\right]$$

$$\implies \left\|\mathbf{A}_{\mathbf{S}^*}\bar{\mathbf{x}} - \mathbf{A}_{\mathbf{S}_k}(\mathbf{A}_{\mathbf{S}_k}^T\mathbf{A}_{\mathbf{S}_k})^{-1}\mathbf{A}_{\mathbf{S}_k}^T\mathbf{A}_{\mathbf{S}^*}\bar{\mathbf{x}}\right\|_2^2 = \frac{n\left[\epsilon^2 + k\epsilon(1-\epsilon)\right]}{(1 + (k-1)(1-\epsilon))^2} \tag{C.35}$$

Using (C.34) and (C.35) we get

$$\frac{1}{n}\left\|\mathbf{A}_{\mathbf{S}_k}\hat{\mathbf{x}}_k^{\text{OMP}} - \mathbf{A}_{\mathbf{S}^*}\bar{\mathbf{x}}\right\|_2^2 = \frac{1}{n}\left\|P(\mathbf{A}_{\mathbf{S}_k})\boldsymbol{\eta}\right\|_2^2 + \frac{1}{n}\frac{n\left[\epsilon^2 + k\epsilon(1-\epsilon)\right]}{(1 + (k-1)(1-\epsilon))^2}$$

$$- \frac{2}{n}\langle P(\mathbf{A}_{\mathbf{S}_k})\boldsymbol{\eta}, (\mathbf{I} - P(\mathbf{A}_{\mathbf{S}_k})\mathbf{A}_{\mathbf{S}^*}\bar{\mathbf{x}})\rangle$$

$$= \frac{1}{n}\left\|P(\mathbf{A}_{\mathbf{S}_k})\boldsymbol{\eta}\right\|_2^2 + \frac{\left[\epsilon^2 + k\epsilon(1-\epsilon)\right]}{(1 + (k-1)(1-\epsilon))^2}$$

$$= \frac{1}{n}\left\|P(\mathbf{A}_{\mathbf{S}_k})\boldsymbol{\eta}\right\|_2^2 + \frac{\epsilon}{1 + (k-1)(1-\epsilon)} \tag{C.36}$$

When $\epsilon = 1 - 4/s^*$ and $s^* \geq 8$, we have

$$\frac{1}{n}\left\|\mathbf{A}_{\mathbf{S}_k}\hat{\mathbf{x}}_k^{\text{OMP}} - \mathbf{A}_{\mathbf{S}^*}\bar{\mathbf{x}}\right\|_2^2 = \frac{1}{n}\left\|P(\mathbf{A}_{\mathbf{S}_k})\boldsymbol{\eta}\right\|_2^2 + \frac{1 - \frac{4}{s^*}}{1 - \frac{4}{s^*} + \frac{4k}{s^*}}$$

$$\geq \frac{\frac{1}{2}}{\frac{1}{2} + \frac{4k}{s^*}}$$

$$\geq \frac{s^*}{9k} \tag{C.37}$$

Since $s \leq \frac{1}{2\sigma}\sqrt{\frac{n}{\log d/\delta}}$,

$$\frac{1}{n}\left\|\mathbf{A}_{\mathbf{S}_k}\hat{\mathbf{x}}_k^{\mathrm{OMP}} - \mathbf{A}_{\mathbf{S}^*}\bar{\mathbf{x}}\right\|_2^2 \geq \frac{2\sigma s^*}{9}\sqrt{\frac{\log d/\delta}{n}} \tag{C.38}$$

$$\geq \sigma\sqrt{\frac{\log d/\delta}{n}}$$

$$\geq \sigma\frac{\sqrt{\log d/\delta}}{n}\sqrt{n}$$

$$\geq \frac{2\sigma^2 s \log (d/\delta)}{n}$$

$$\geq \frac{\sigma^2 \widetilde{\kappa}_s s^* \log (d/\delta)}{18n}$$

In the last line, we use $s \geq \frac{1}{36}\widetilde{\kappa}_s s^*$.

$\square$

**Lemma C.1.** *Suppose the same conditions as Theorem 4.3 hold. For the lower bound matrix defined in the proof of 4.3, suppose that a correct support has not been recovered until step $T \leq s$, then for all $k \in \mathbf{S}_t$, $\mathbf{A}_k^T\boldsymbol{\eta} \geq 0$.*

*Proof.* With probability $1 - \delta$, for all $i \in [d]$, $\left|\mathbf{A}_i^T\boldsymbol{\eta}\right| \leq \sigma\sqrt{2n\log\frac{d}{\delta}}$. Thus, for iteration $k + 1$,

$$\sum_{l \in \mathbf{S}_k}\mathbf{A}_l^T\boldsymbol{\eta} \leq k\sigma\sqrt{2n\log\frac{d}{\delta}}$$

$$\leq s\sigma\sqrt{2n\log\frac{d}{\delta}}$$

$$\leq \left(\sqrt{\frac{n}{32\sigma^2\log\frac{d}{\delta}}}\right)\sigma\sqrt{2n\log\frac{d}{\delta}}$$

$$\leq \frac{n}{4}$$

Further, since $s^* \geq 8$, $\epsilon \geq 2/3 \geq \sqrt{1-\epsilon}$. Thus, $\left[n\epsilon - \sqrt{1-\epsilon}\sum_{l \in \mathbf{S}_k}\mathbf{A}_l^T\boldsymbol{\eta}\right] \geq 3n\epsilon/4 \geq n/2$. Now,

$$\frac{\sqrt{1-\epsilon}}{1 + (k-1)(1-\epsilon)} \geq \frac{\sqrt{1-\epsilon}}{1 + (s-1)(1-\epsilon)}$$

$$\geq \frac{\sqrt{1-\epsilon}}{2s(1-\epsilon)}$$

$$\geq \frac{1}{2s}$$

$$\implies \frac{\sqrt{1-\epsilon}}{1 + (k-1)(1-\epsilon)}\left[n\epsilon - \sqrt{1-\epsilon}\sum_{l \in \mathbf{S}_k}\mathbf{A}_l^T\boldsymbol{\eta}\right] \geq \frac{n}{4s}$$

$$\geq \frac{\sqrt{n}\left(\sqrt{32}\sigma s\sqrt{\log\frac{d}{\delta}}\right)}{4s}$$

$$\geq \sigma\sqrt{2n\log\frac{d}{\delta}}$$

$$\geq \left|\mathbf{A}_j^T\boldsymbol{\eta}\right| \geq 0 \qquad (\forall j \in (\mathbf{S}^* \cup \mathbf{S}_{k-1})^{\mathrm{c}})$$
$$\tag{C.39}$$

From C.23, if a $j \in (\mathbf{S}^* \cup \mathbf{S}_{k-1})^{\mathsf{c}}$ is picked, then it maximizes:

$$\left| \frac{\sqrt{1-\epsilon}}{1 + (k-1)(1-\epsilon)} \left[ n\epsilon - \sqrt{1-\epsilon} \sum_{l \in \mathbf{S}_k} \mathbf{A}_l^T \eta \right] + \mathbf{A}_j^T \eta \right|$$

From C.39 we can see that for any $j, j'$ such that $\mathbf{A}_j^T \eta \geq 0$ and $\mathbf{A}_{j'}^{\prime T} \eta \leq 0$,

$$\left| \frac{\sqrt{1-\epsilon}}{1 + (k-1)(1-\epsilon)} \left[ n\epsilon - \sqrt{1-\epsilon} \sum_{l \in \mathbf{S}_k} \mathbf{A}_l^T \eta \right] + \mathbf{A}_j^T \eta \right|$$

$$= \frac{\sqrt{1-\epsilon}}{1 + (k-1)(1-\epsilon)} \left[ n\epsilon - \sqrt{1-\epsilon} \sum_{l \in \mathbf{S}_k} \mathbf{A}_l^T \eta \right] + \mathbf{A}_j^T \eta$$

$$\geq \frac{\sqrt{1-\epsilon}}{1 + (k-1)(1-\epsilon)} \left[ n\epsilon - \sqrt{1-\epsilon} \sum_{l \in \mathbf{S}_k} \mathbf{A}_l^T \eta \right]$$

$$\geq \frac{\sqrt{1-\epsilon}}{1 + (k-1)(1-\epsilon)} \left[ n\epsilon - \sqrt{1-\epsilon} \sum_{l \in \mathbf{S}_k} \mathbf{A}_l^T \eta \right] + \mathbf{A}_{j'}^{\prime T} \eta$$

Thus, OMP will indeed pick up an index $j$ such that $\mathbf{A}_j^T \eta \geq 0$. This proof holds for all steps $k < T$. Thus we can say that for all $k \leq T$, the index that is picked up (say $\mathbf{A}_l$) satisfies $\mathbf{A}_l^T \eta \geq 0$. $\qquad \square$