[Reviews · NeurIPS 2018]

Reviewer 1



OMP is a well known and widely used method for sparse learning from linear regression. There have several works which provide the support recovery of OMP under different settings, e.g., noiseless setting under various conditions like incoherence. In the noisy setting, the existing bound in [28] does not match the known lower bounds. The main contribution of this work is to improve these support recovery results for OMP and obtain matching lower bounds. Generally, this work is very well written. The studied problem about the support recovery of OMP is important and of great interest. The authors provide sufficient theoretical results for OMP. These results extend the applicability of OMP. For the experiments, how to initialize the solution? -------------------------------------- Updates after author feedback the authors have addressed my concern. I increase my rating from 6 to 7.

Reviewer 2



##Summary## The paper studies the support recovery performance of orthogonal matching pursuit, where the goal is to estimate the position of the non-zero elements of a high-dimensional signal from its compressed measurements. While results under the L_2 metric are well-known, and sharp RIP conditions have been established, there are still some theoretical gaps in understanding support recovery: what is the sharp condition? The work follows this line and present lower and upper bounds under natural assumprtions. ##Detailed Comments## I am satisfied with most of the discussion in the paper. Below are a few comments that I hope the authors could address in the feedback. - I am curious how did the authors obtain better dependence on the condition number than [20,21,25]. Is it possible to sketch the main technique/idea right after Remark 3? - The relaxed sparsity idea for OMP was used in [28]. What is the major difference between Theorem 3.1 and [28]? - line 244: I did not really follow why noise helps recovery. In 1-bit matrix completion, noise does help distinguish the model. But what is the intuition here? - The equation numbering is way strange. Authors need to improve the typeset in the revision. Authors should also keep the style of bib entries consistent. ---Updates After Author Feedback--- Authors addressed my concerns and I feel this is a nice work. In particular, authors draw more careful analysis on the progress of OMP, and tightened the old bounds [28] that have been used for a decade. Personally I like their theoretical results. I hope they will sketch the proof technique and add more comparison to [20,21,25,28] in the main body, say list the bounds in a table. I believe it is a significant contribution for the community, and I increase my rating from "7" to "8".

Reviewer 3



The paper provides an analysis of OMP that shows lower and upper bounds for sparse recovery in terms of the sparsity and matrix conditioning, where the lower bounds correspond to specific matrix constructions and where the two bounds are of the same order. While the results are interesting, there are some issues with the description and narrative that need to be addressed to recommend acceptance. The paper describes distributions for A and y starting at page 3 line 108, but a distribution is only specified within Section 3.1; the former may be just a wording issue. The practical impact of Theorems 3.1-3.3 appears limited due to the very small upper bound on delta (which results on strong requirements on xmin and s+s*) and the very large value of the constant C. The authors should contrast the order relationships implied by the theorems (and the improvements with respect to the literature) with the numerical results to highlight the value of the result not being lost by the constant. There is no numerical evaluation of the upper bounds of Theorems 3.1-3.3. Some of the derivations in page 5 were not clear to me (3.0.4, 3.0.9). Minor comments follow. Page 3, Defs. 2.1-2.2: Why not just write w = z-x? This way these conditions more closely resemble the asymmetric restricted isometry property. Page 3, Line 100: From your reply, I would consider C a parameter as well. Page 6, Line 190: It is not clear what "the support is distributed evenly" means - you should make it explicit as in your response.